# Unconventionally fast transport through sliding dynamics of rodlike particles in macromolecular networks

Xuanyu Zhang[1,2,5], Xiaobin Dai[1,2,5], Md Ahsan Habib [3,5], Lijuan Gao[1,2], Wenlong Chen[1,2], Wenjie Wei[1,2], Zhongqiu Tang[3], Xianyu Qi[4], Xiangjun Gong[4], Lingxiang Jiang [3] ✉ & Li-Tang Yan [1,2] ✉

Transport of rodlike particles in confinement environments of macromolecular networks plays crucial roles in many important biological processes and technological applications. The relevant understanding has been limited to thin rods with diameter much smaller than network mesh size, although the opposite case, of which the dynamical behaviors and underlying physical mechanisms remain unclear, is ubiquitous. Here, we solve this issue by combining experiments, simulations and theory. We find a nonmonotonic dependence of translational diffusion on rod length, characterized by length commensuration-governed unconventionally fast dynamics which is in striking contrast to the monotonic dependence for thin rods. Our results clarify that such a fast diffusion of thick rods with length of integral multiple of mesh size follows sliding dynamics and demonstrate it to be anomalous yet Brownian. Moreover, good agreement between theoretical analysis and simulations corroborates that the sliding dynamics is an intermediate regime between hopping and Brownian dynamics, and provides a mechanistic interpretation based on the rod-length dependent entropic free energy barrier. The findings yield a principle, that is, length commensuration, for optimal design of rodlike particles with highly efficient transport in confined environments of macromolecular networks, and might enrich the physics of the diffusion dynamics in heterogeneous media.

The transport of guest particles in confinement environments of macromolecular networks holds significant importance in various fields, including biological science, materials science, and soft matter physics[1–8]. It serves as a fundamental problem that underlies the behavior of diverse systems. For instance, in biological systems, the selective permeation of bacteria or antibiotics across mucus is influenced by the transport dynamics occurring in confined spaces[1–3].

Besides, it determines the organization of nanoparticles within polymer-network-based nanocomposites, enabling the development of advanced materials with characteristic properties[4,5]. Developing efficient drug delivery systems that target tissues covered with extracellular matrix also relies on understanding and manipulating guest particle transport in such environments[6–8]. Thus, understanding the dynamics of particle transport is essential for advancements in

[1]State Key Laboratory of Chemical Engineering, Department of Chemical Engineering, Tsinghua University, 100084 Beijing, China. [2]Key Laboratory of Advanced Materials (MOE), Tsinghua University, 100084 Beijing, China. [3]South China Advanced Institute for Soft Matter Science and Technology, School of Emergent Soft Matter, South China University of Technology, 510640 Guangzhou, China. [4]Faculty of Materials Science and Engineering, South China University of Technology, 510640 Guangzhou, China. [5]These authors contributed equally: Xuanyu Zhang, Xiaobin Dai, Md Ahsan Habib. ✉e-mail: jianglx@scut.edu.cn; ltyan@mail.tsinghua.edu.cn

nanocomposite design, medical treatments, and drug delivery strategies with improved efficiency and specificity. Physically, the dynamics is governed by the corresponding effective free energy landscape contributed by the interaction between particle and networks strands as well as the elastic deformation of network strands[5,9–13]. For spherical particles, the interplay of these two factors strongly depends on the size ratio between the network mesh and particles, resulting in various diffusion regimes including Brownian, hopping and trapped dynamics[10–13]. However, this simple picture breaks down for rodlike particles because anisotropic shape gives rise to additional competing length scales[14–17]. As a matter of fact, numerous particles in biological systems, such as bacteria, viruses, nucleic acids, proteins, and polypeptides, hold rodlike geometry[18]. As various length scales associated with the structure significantly increase the complexity of the dynamics, the fundamental dynamics of such systems is far from established.

In most studies on the transport of rodlike particles in macromolecular networks, rod diameters are much smaller than typical length scales of network mesh[19–24]; the elastic energy due to strand deformation thereby plays a minor role in the effective free energy landscape. However, the opposite limit, with rod diameter comparable with or even larger than the latter, is ubiquitous. For example, to demonstrate this point, we perform a statistical analysis of the ratio of diameter $d$ of rodlike bacteria to averaged mesh size $a_x$, $d/a_x$, for many experiments of typical bacteria in different types of mucus consisting of biomacromolecular networks[24–34]. As demonstrated in Fig. S1 and listed in Table S1, surprisingly, $d/a_x$ ranges from about 0.8 to almost 2.0. In this case, the entropic free energy barrier due to the conformational penalty of strands deformed by particles may overwhelm the interactions, leading to new dynamical regimes[35]. Unfortunately, little is known about the diffusion mechanisms regarding the transport of such *thick* rodlike particles in macro-molecular networks, leaving an urgent and critical issue to be addressed.

Here, by combining experiment, simulation and theory, we provide, for the first time, the systematical exploration of the dynamics of rodlike particles of diameters comparable to mesh size in a cross-linked macromolecular network, in order to unravel the fundamental mechanisms underlying the transport in the essential systems. The merits of the results are listed as follows: (1) It reports the nonmonotonic dependence of translational diffusion on rod length, where exceptional fast dynamics occurs once rod length reaches around an integral multiple of network mesh size, in stark contrast to the monotonic dependence reported previously for thin rods[20]. (2) It not only reveals that the fast transport follows sliding dynamics but also gives the analytical expressions of the time-displacement distribution of sliding dynamics, clarifying its physical relationship with hopping and Brownian dynamics. (3) This work provides new principles for the optimal design of particle transport in various networks, biological or synthetic.

## Results
### Single-particle tracking of rods in synthetic networks
We begin by examining the diffusion behavior of rods with different lengths in a synthetic macromolecular network, where the diameters of rods $d$ are comparable with the mesh size $a_x$ of the network. The experimental network is polyethylene glycol diacrylate (PEGDA) network, which possesses excellent biocompatibility[36]. The PEGDA network is synthesized under ultraviolet (UV) irradiation (see "Methods" section and Supplementary Information I for more details)[37]. $a_x$ of this network is estimated to be $21.0 \pm 1.8$ nm (Table S2)[38,39]. The experimental rods are PEG-capped Au nanorods (Au-NRs), with lengths $L$ ranging from $30.6 \pm 3.4$ nm to $61.4 \pm 7.7$ nm and diameters, including grafted-layer thickness, of around 19.1 nm. Rod sizes are measured from TEM images (Fig. S2), and the averaged values are listed in Table S3. The images of trajectories are taken by using dark field techniques, as schemed in Fig. 1a[40–43]. Particularly, in the confinement environment due to the

scales of such large diameter and length of the rods considered in the current work, the waiting time for a rotating event possesses a period much larger than the observation time, underscoring that the rotational dynamics plays a trivial role in the transport of rods in the experiments (see Supplementary Information V for more details). Thus, in the following discussion, we focus on the transitional dynamics of the rods.

To detect the particles in the image stack, the Crocker-Grier algorithm is applied in the experiments[44,45]. Representative snapshots of rods are illustrated in Fig. 1b, where the initial and spontaneous positions of the rod centers are marked by the dashed cyan and magenta lines, respectively. Through measuring the displacement between magenta and cyan dashed lines, it can be found that the rod with $L/a_x = 2.63 \pm 0.31$ manifests the discontinuous "hopping" motion punctuated by waiting periods; that is, the rod remains almost stationary for ~300 s and then hops to a new position. However, the rod with $L/a_x = 2.97 \pm 0.35$ exhibits the continuous diffusion, yielding an exceptional situation where different dynamic behaviors appear. Strikingly, such a change of the dependence of diffusivity on rod length for the thick rod is distinct from the monotonic situation for the thin rod, as reported in previous works[20]. This can be confirmed through examining the trajectories obtained by Nearest Neighbor Search tracker[45] (Fig. S3). Figure 1c presents the representative trajectories for rods with different lengths corresponding to the hop and continuous motions. Specifically, one can identify that the trajectories of the rods with $L/a_x = 2.02 \pm 0.18$ and $L/a_x = 2.97 \pm 0.35$ undergo remarkably faster diffusion than that with $2.51 \pm 0.27$ where the "hop" can be definitely observed, indicating unconventional dynamics of the thick rods with lengths of integral multiple mesh size.

For a more quantitative analysis, we first examine the mean square displacement (MSD), $\langle \Delta z^2(t) \rangle = \langle [z(t + t_0) - z(t_0)]^2 \rangle$, where $z(t)$ is the displacement of the center of mass along the direction of the rod contour that is determined by the eigendecomposition algorithm[46] (see section I of the Supplementary Information). The typical MSDs for a set of $L/a_x$, at $d/a_x = 1.0 \pm 0.1$ are shown in Figs. 1d and S4[47]. Indeed, the diffusivity of rods significantly depends on the rod length for these thick rods. It can be identified that, when the rod length $L$ reaches around an integral multiple of the network mesh size $a_x$, i.e., $L/a_x = 2.02 \pm 0.18$ and $2.97 \pm 0.35$, they exhibit a faster diffusion than the rods with lengths noncommensurate with the mesh size, i.e., $L/a_x = 1.46 \pm 0.16$, $2.51 \pm 0.27$ and $2.63 \pm 0.31$. This indicates that the unconventionally fast dynamics occurs upon that $L$ reaches around an integral multiple of $a_x$, that is, a commensurate length. To further examine this length-dependent fast diffusive behavior, we calculate the $D/D_0$, i.e., the ratio between the diffusion coefficients in the network and neat solvent (Fig. 1e). Indeed, $D/D_0$ reaches a maximum for a rod with commensurate length, clarifying this unique dynamical behavior. Furthermore, we calculate the displacement probability distribution function (DPDF) $G_s(z, t)$ corresponding to the diffusion dynamics of noncommensurate ($L/a_x = 2.51 \pm 0.27$) and commensurate ($L/a_x = 2.97 \pm 0.35$) rods, as shown in Figs. 1f and S5 and Figs. 1g and S6, respectively. For the noncommensurate rod, $G_s(z, t)$ lines exhibit regular peaks, indicating that the rod transport undergoes hopping-like dynamics. In contrast, $G_s(z, t)$ lines of the commensurate rod become shallow and irregular peaks. Such an exceptional dependence of the dynamics clarifies that the length commensuration is critical for the dynamical behaviors of thick rods, yielding new principles for the optimal design of rodlike particles with highly efficient transport in macromolecular networks. For instance, as the networks and surface chemistry of rods used in the experiments possess excellent biocompatibility, it can be anticipated that the findings provide useful guidelines for designing the shape of drug delivery systems[36,48].

### Detailed microscopic dynamics revealed by molecular simulations
To elucidate the physical origin of the unconventionally fast dynamics, we turn to the detailed microscopic dynamics of thick rods in

macromolecular networks. We simulate the transport of a rodlike particle in a cross-linked network using dissipative particle dynamics (DPD)[49]. Full technical details on the simulation model are described in the Methods and Supplementary Information II and briefly introduced here. The configuration of the network is taken to be a hexa-functional network[50], as illustrated in Fig. 2a. A general particle-building model[51] is adopted to build a set of rod particles fabricated by numbers of beads, which move as rigid bodies[52]. To bring out the entropic nature of the interplay between rods and network strands, the rod-strand interaction is set to be the same as that between like beads, capturing the physical nature of the interaction in this system where the free energy change is predominately contributed by the entropy[53]. For all systems, the average network mesh size is at around $a_x = 3.35r_c$, while various values of $d$ and $L$ are set to consider their roles in the rod dynamics. $r_c$ is the cutoff radius of the soft potentials and is used as the length unit of

the system. Particularly, $d$ is set to be comparable to the mesh size, ranging from $1.3a_x$ to $1.9a_x$, and $L$ ranges from $1.5a_x$ to $4.4a_x$ to avoid the effect of rod rotation. The normalized sizes $d/a_x$ and $L/a_x$ are used, representing the size matching between a rod and a network mesh. Typical MSDs for a set of $L/a_x$ at $d/a_x = 1.4$ are shown in Fig. 2b. At short time scales, all rods move ballistically and $\langle \Delta z^2(t) \rangle$ scales as $t^2$. At longer time scales, for rods with a length noncommensurate with $L/a_x = 1.5$, 2.6 and 3.5, a plateau apparently emerges between the short- and long-time diffusion regimes, indicating that the rod transport undergoes slow hopping-like dynamics[10]. In striking contrast, at $L/a_x = 2.1$, 3.1 and 4.1, they evolve directly from the ballistic regime to the Fickian regime with $\langle \Delta z^2(t) \rangle \sim t$ demonstrating that fast transport dynamics occurs upon that $L$ reaches around an integral multiple of $a_x$. The diffusion coefficients corresponding to the MSDs are illustrated in Fig. 2c, which also demonstrates the nonmonotonic dependence of the diffusivity on

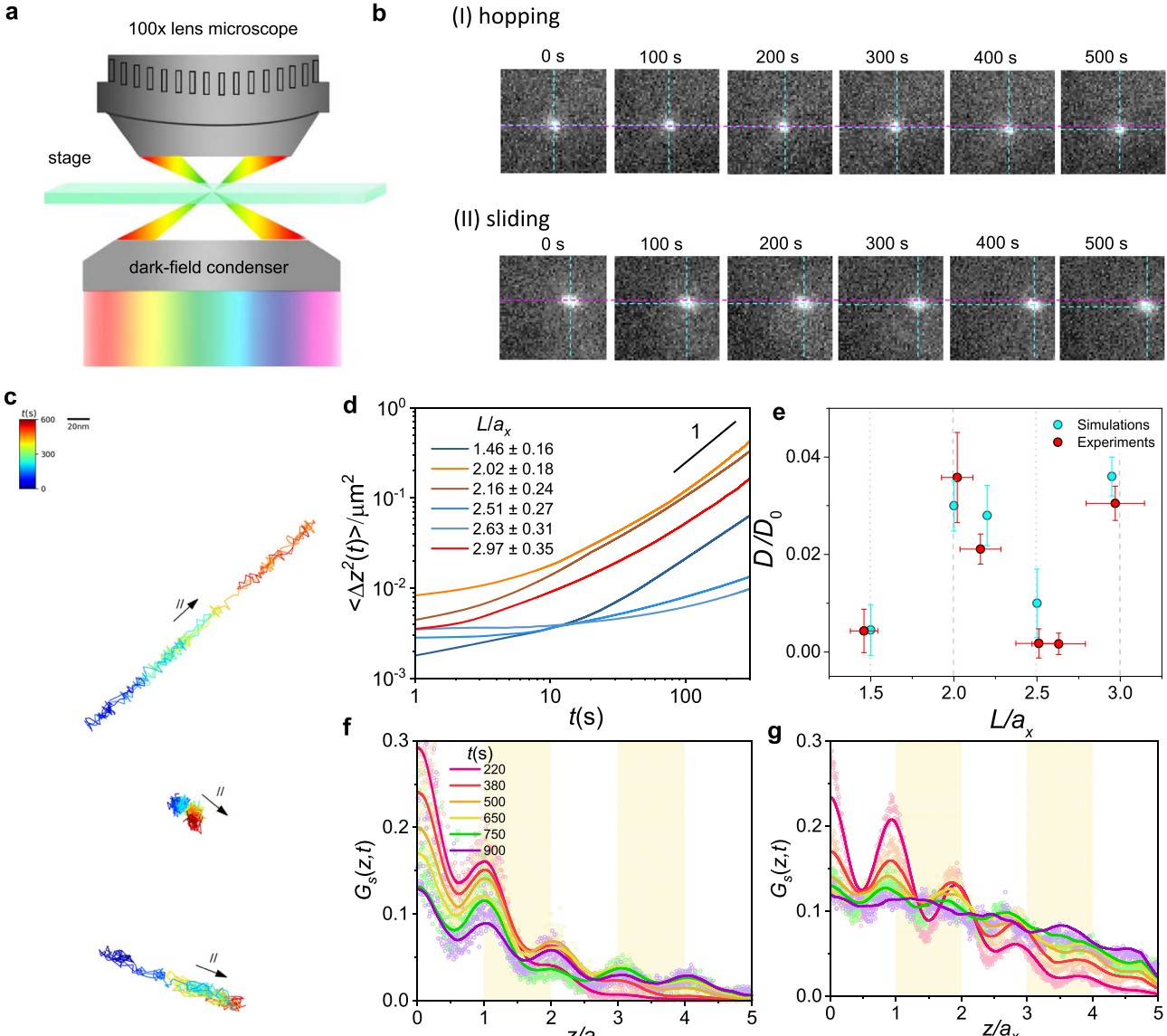

**Fig. 1 | Single-particle tracking of rods in synthetic networks. a** Schematic illustration of dark-field microscopy (DFM) technique to track the nanorod diffusion. **b** Representative time series of snapshots of nanorods diffusing in PEGDA network, where $L/a_x$: (I) $2.51 \pm 0.27$ and (II) $2.97 \pm 0.35$. The edge length of each snapshot is 0.5 μm. The cyan dashed lines mark the spontaneous position of the center of the nanorod, while the magenta dashed line denotes the initial position of the rod center ($t = 0$ s). **c** Typical trajectories of rods with different $L/a_x$: $2.02 \pm 0.18$, $2.51 \pm 0.27$ and $2.97 \pm 0.35$ from top to the bottom, corresponding respectively to the fast, hopping and fast dynamics. The color and scale bars on the left top indicate the values of the temporal and spatial scales of the trajectories. **d** $\langle \Delta z^2(t) \rangle$ for different $L/a_x$ in experiments. **e** $D/D_0$ as a function of $L/a_x$. The error bars represent the standard deviation. Experimental $G_s(z, t)$ of rods with **f.** $L/a_x = 2.51 \pm 0.27$ and **g.** $L/a_x = 2.97 \pm 0.35$, where solid lines represent smoothed results of circles.

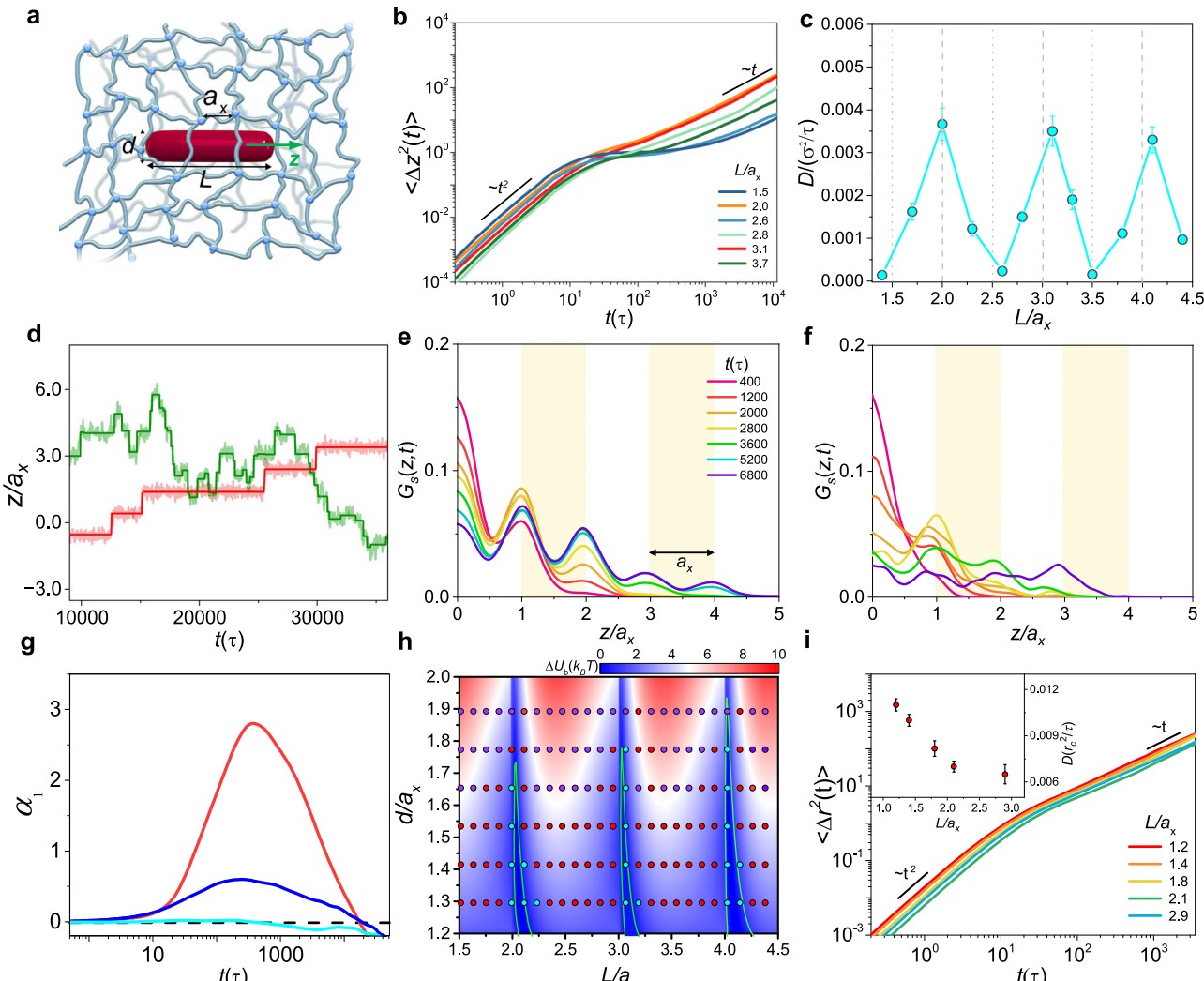

**Fig. 2 | Detailed microscopic dynamics revealed by molecular simulations.**
**a** Schematic of a rodlike particle of diameter $d$ and length $L$ in a macromolecular network with mesh size $a_x$. **b** $\langle\Delta z^2(t)\rangle$ for different $L/a_x$ at $d/a_x = 1.4$. **c** $D$ for different $L/a_x$ at $d/a_x = 1.4$. **d** Typical trajectories of the center of mass of the corresponding rods with $L/a_x = 2.6$ (red) and $L/a_x = 3.1$ (green) at $d/a_x = 1.4$ along $z$-axis, where thin lines represent unsmoothed original displacement while thick lines represent smoothed displacement determined by a wavelet-based method[71]. Simulated $G_s(z, t)$ of rods with (**e**) $L/a_x = 2.6$ and (**f**) $L/a_x = 3.1$ at $d/a_x = 1.4$ along $z$-axis, showing

hopping-type diffusion and fast diffusion respectively. **g** $\alpha_1$ for Brownian dynamics (cyan), fast dynamics (blue) and hopping dynamics (red). **h** Diagram of rod dynamics interrelating to $L/a_x$ and $d/a_x$. Circles: simulation results of (red) hopping, (cyan) fast, and (purple) trapped diffusion behaviors. The contour map: theoretical results of $U_b$ with various $d/a_x$ and $L/a_x$. The green lines: boundary (where $U_b = k_BT$) between the fast and hopping regimes, determined by a theoretical model. The color bar at the upper right corner indicates the value of $U_b$. **i** $\langle\Delta z^2(t)\rangle$ and $D$ for different $L/a_x$ at $d/a_x = 0.18$.

$L$. This yields optimal values for the fast diffusion of such thick rods in macromolecular networks.

In order to provide a detailed insight into the fast transport mentioned above, we compare dynamical behaviors in the regimes of hopping and fast dynamics through calculating different parameters, i.e., typical trajectories in the direction of $z$-axis (Fig. 2d) and the corresponding $G_s(z, t)$ (Fig. 2e, f). The trajectory of the rod in the regime of hopping dynamics shows that the rod undergoes constrained motion punctuated by significant large-scale full hops (the red line in Fig. 2d). This can be confirmed from the $G_s(z, t)$ lines where regular peaks emerge and the distance between adjacent peaks is around $1.0a_x$, indicating that the rod hops from a network cell to its neighboring one (Fig. 2e). By contrast, in the regime of fast dynamics, the rod that experiences the fast transport dynamics does not exhibit hopping-type events as sharp as those observed for the noncommensurate rods. Instead, only random and local waiting intervals of small scales appear in the trajectory (the green line in Fig. 2d), resulting in shallow and irregular peaks in lines of $G_s(z, t)$ (Fig. 2f). By comparing Fig. 2e, f to

Fig. 1f, g respectively, a good agreement between simulation and experimental results is identified, corroborating the hopping and fast dynamics for the noncommensurate and commensurate rods. Clearly, the $G_s(z, t)$ for the commensurate rod is non-Gaussian, but is different from those of hopping dynamics with regular peaks or Brownian dynamics with Gaussian distribution (Fig. S7). For the purpose of evaluating the non-Gaussianity, we further calculate the corresponding non-Gaussian parameter $\alpha_1(t) = (1/3)\langle\Delta z^4(t)\rangle/\langle\Delta z^2(t)\rangle^2 - 1$ in the direction of $z$-axis as displayed in Fig. 2g. Indeed, the dynamics in the fast regime presents the weak non-Gaussianity intermediating between the strong dynamical heterogeneity of hopping diffusion and the Gaussian behavior of Brownian dynamics.

To evaluate the generality of such dynamic dependence on the length scales of rod and network, we systematically explore the diffusion behaviors of thick rods as a function of $L/a_x$ and $d/a_x$, allowing us to construct a diagram of diffusion dynamics in the two-parameter space, as depicted in Fig. 2h. It can be found that for each size ratio interval $L/a_x \in [n, n+1]$ with $n = 0, 1, 2, \ldots$, the diffusion of rod possesses

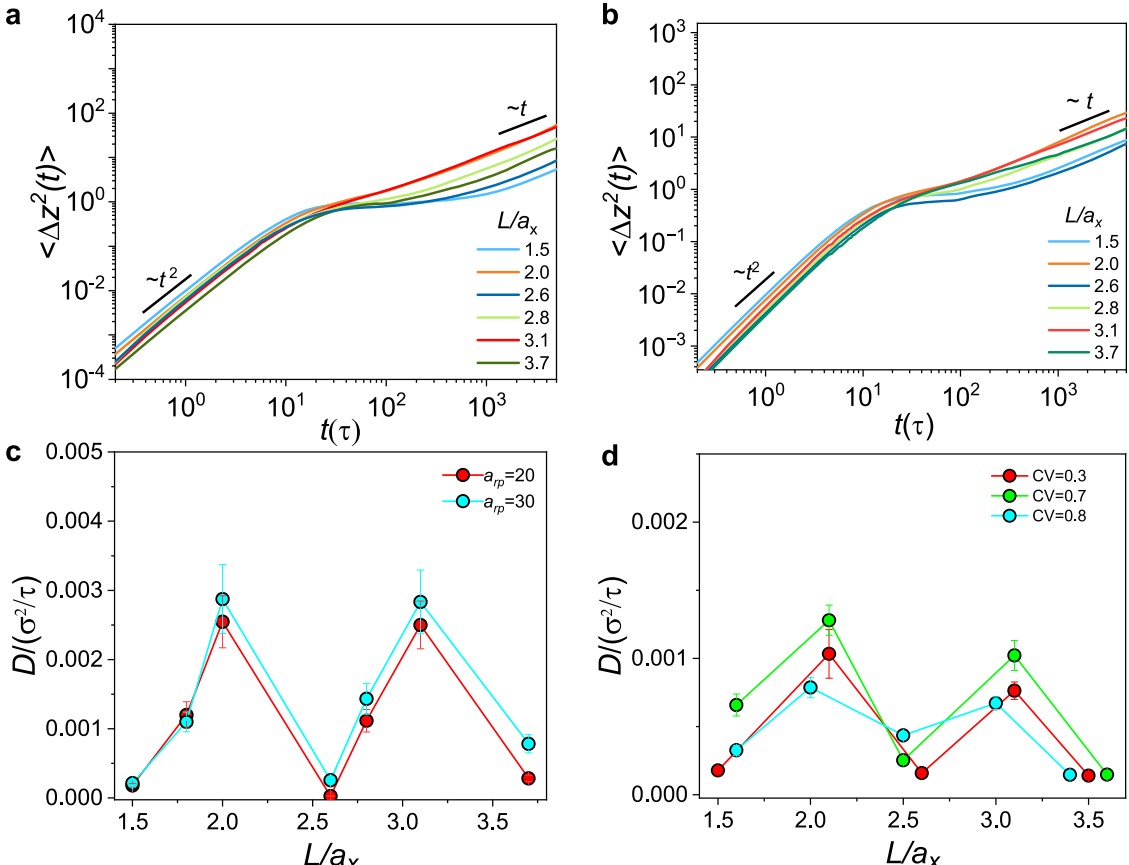

**Fig. 3 | Entropic effect examined through evaluating diverse factors. a** $\langle\Delta z^2(t)\rangle$ for different $L/a_x$ at $d/a_x = 1.4$ when $a_{rp} = 20$. **b** $\langle\Delta z^2(t)\rangle$ for different $L/a_x$ at $d/a_x = 1.4$ and CV = 0.3. **c** $D$ for different $L/a_x$ at $d/a_x = 1.4$ when $a_{rp} = 20$ (red) and 30 (blue). **d** $D$ for different $L/a_x$ at $d/a_x = 1.4$, CV = 0.3 (red); $d/a_x = 1.4$, CV = 0.7 (green); and $d/a_x = 1.5$, CV = 0.8 (blue).

similar dynamic types. That is, three characteristic regimes can be discriminated as denoted by the colored circles in Fig. 2h: when $L$ is noncommensurate with $a_x$, the rod diffusion is featured by hopping between neighboring network cells for small $d/a_x$ while it turns to the trapped dynamics[10] characterized by slight fluctuation around its equilibrium position upon about $d/a_x > 1.6$; interestingly, between two neighboring hopping regimes there indeed exists the regime of faster longitudinal dynamics, where $L$ reaches around an integral multiple of $a_x$, corresponding to Fig. 2b. The result in the diagram clearly corroborates the nonmonotonic dependence of the diffusivity on $L$ for thick rods as well as the regime of the unconventionally fast dynamics. The more random fluctuation in the displacement of the faster dynamics implies that the rod undergoes a lower free energy barrier, and the thermal noise as well as the strong effect of the local environment may thereby play a nontrivial role.

**Entropic effect examined through evaluating diverse factors**
In order to explore the physical mechanism underlying the non-monotonic dependence of the diffusivity of such thick rods, we conduct comprehensive investigations into the influence of diverse factors attributed to the large diameters of rods. To this end, first, the diameter is reduced and thereby representative thin rods, with $d/a_x = 0.18$, are considered. Figure 2i illustrates the dependence of MSDs and diffusion coefficients on the length of the thin rods. The simulation results demonstrate that increasing the rod length monotonically reduces the diffusion coefficient, which is consistent with some previous studies[20] but in sharp contrast to the nonmonotonically decreased diffusivity with respect to the increase of rod length for thick rods (Fig. 2c). This implies that the entropic contribution,

originated from the conformational penalty of loops, which consist of a group of end-to-end connected polymer strands[9], deformed by the thick rods with large diameter, plays a key role in the free energy barrier. Basically, the number of loops around a thick rod is determined by the rod length, leading to the rod-length-dependent free energy barrier for thick rods.

Next, to further examine the aforementioned entropic nature of the length-dependent, nonmonotonic diffusion dynamics of thick rods, we evaluate the influence of the enthalpy contributions through performing simulations by fixing $d/a_x$ and varying the particle-strand interaction parameter, $a_{rp}$, which measures the relative strength of the interaction between rod and network strand (see Supplementary Information II for more details). A larger $a_{rp}$ corresponds to a stronger repulsion between these both species, while between like species it is chosen to be 25[49]. As shown in Figs. 3a, c and S8, the faster longitudinal dynamics continues once the rod length reaches around an integral multiple of $a_x$ for both attractive ($a_{rp} = 20$) and repulsive ($a_{rp} = 30$) particle-strand interactions, resembling the above results for the systems governed by almost purely entropic effects. It underscores that the entropic contribution still dominates the behaviors over a wide range of the energetic interactions.

Last, we quantify the distribution of mesh sizes for some typical biological networks and thereby examine the entropic effect regarding the fast dynamics of thick rods in them, in view of the fact that there is a distribution of mesh sizes in a real network which causes the poly-dispersity of molecular structures of the networks. Specifically, the distribution of mesh sizes is quantified through calculating the coefficient of variation of mesh sizes, defined as CV = $\sigma_{ax}/a_x$, where $\sigma_{ax}$ is the standard deviation of mesh sizes. The CV of some typical biological

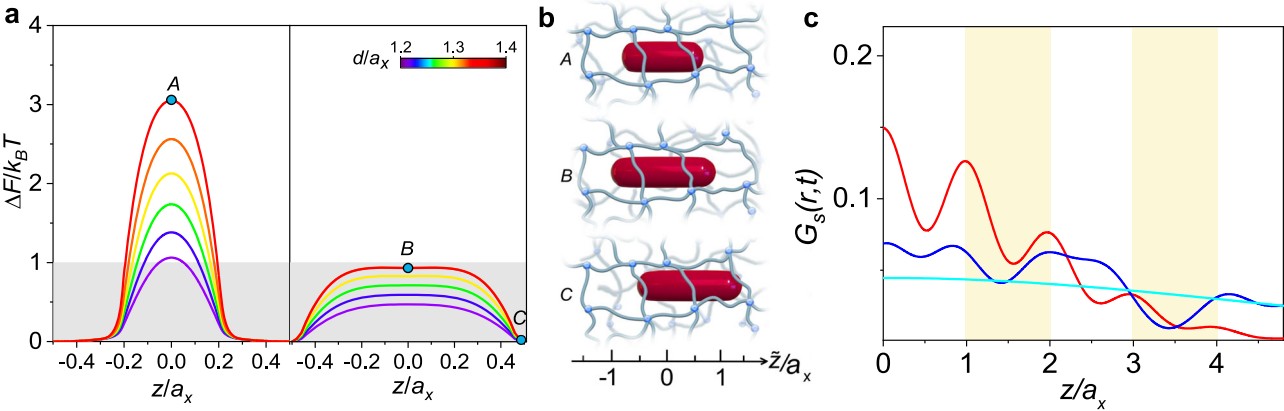

**Fig. 4 | Physical origin established by theoretical analysis. a** $\Delta F$ as a function of $\tilde{z}$ for various $d/a_x$, where $L/a_x = 1.5$ (left) and 2.0 (right). The color bar indicates the value of $d/a_x$. Typical positions of rod center for various $L/a_x$ are schematically illustrated in **b**. A. $\tilde{z}/a_x = 0$, $L/a_x = 1.5$; B. $\tilde{z}/a_x = 0$, $L/a_x = 2.0$; C. $\tilde{z}/a_x = 0.5$, $L/a_x = 2.0$. **c** Theoretical result, of $G_s(\tilde{z}, t)$ for (red) hopping, (blue) sliding and (cyan) Brownian dynamics.

mucus approximately ranges from 0.1 to 0.8[23,54–56], as listed in Table S4. Thus, we perform simulations to study the thick rod diffusion in the macromolecular network with the similar distribution of mesh sizes. Based on the analysis of CV in mucus, we set CV = 0.3, 0.7 and 0.8 by varying the distribution of strand lengths in our simulations. As shown in Fig. 3b, d and S9, for $d/a_x = 1.4$, the fast dynamics indeed occurs when the commensurate rods at CV = 0.3 and 0.7. Even at CV = 0.8, the fast dynamics emerges for the thicker rods with $d/a_x = 1.5$, indicating that the thicker the rod, the wider pore size distribution permitted. Nevertheless, the simulation results of the macromolecular networks, whose polydispersity approximates to those of typical biological networks, can basically fall to the physical principles revealed based on the regular network, underscoring that the neat model can be applied to mimic the dynamical behaviors in mucus, especially for the thick rods concerned in our work. Moreover, this indicates again that the entropic effect induced by the thick rods accounts for the unconventional dynamic behaviors, as the influence of the mesh polydispersity is significantly impaired upon the remarkable structural deformation.

## Physical origin established by theoretical analysis

To further confirm the entropic nature and thereby pinpoint the physical origin behind the unconventionally fast transport of a rod in a macromolecule network, we develop a theoretical model to analyze the free energy landscape and dynamical regimes depending on the length scales of rod and network. More details regarding this theoretical model can be found in Supplementary Information III. Considering a canonical ensemble of a rod in a macromolecule network, it is specified by (1) the set of crosslinks $k = \{\mathbf{r}_i\}_{i=1}^M$, with $M$ cross-links between the efficiently bridged Gaussian chains; (2) the collection of linker connections marked as the tuple $(i, j)$; (3) the continuous curve path of linked strands $\mathbf{R}_{ij}(s)$ with contour variable $s \in [0,1]$. For a Gaussian chain of $N$ bonds of Kuhn length $b$, the network mesh size[10], $a_x = bN^{1/2} = 1$ is the unit length of the system. Through coupling the particle effect into the renowned theory of network elasticity[57,58], the partition function of the rod-network system takes the form:

$$Z(\mathbf{r}_{\rm rod}, \mathbf{l}_{\rm rod}) = \prod_k \int d\mathbf{r}_k \prod_{(i,j)} \int D\mathbf{R}_{ij} \delta(\mathbf{r}_i - \mathbf{R}_{ij}(0)) \delta(\mathbf{r}_j - \mathbf{R}_{ij}(1))$$
$$\exp\left[ -\sum_{(i,j)} \int_0^1 \frac{3}{2Nb^2} ||\mathbf{u}_{ij}||^2 + U_{mr}(\mathbf{R}_{ij}, \mathbf{r}_{\rm rod}, \mathbf{l}_{\rm rod}) \right] \quad (1)$$

where $\mathbf{u}_{ij} = d\mathbf{R}_{ij}/ds$ is the tangent direction of a network strand, $\mathbf{r}_{\rm rod}$ and $\mathbf{l}_{\rm rod}$ give the position and direction of the rod, $\delta$ is the delta function, $\beta = 1/k_BT$, is the Boltzmann constant, and $T$ is temperature.

$U_{mr}$ represents the hard-core monomer-rod interaction, which takes the form,

$$U_{mr}(\mathbf{r}_{ij}, \mathbf{r}_{\rm rod}, \mathbf{l}_{\rm rod}) = \begin{cases} \infty & ||\mathbf{r}'_{ij} \cdot \mathbf{l}_{\rm rod}|| < L/2, ||\mathbf{r}'_{ij} - \mathbf{r}'_{ij} \cdot \mathbf{l}_{\rm rod}|| < L/2 \\ 0 & else \end{cases} \quad (2)$$

where $\mathbf{r}'_{ij} = \mathbf{r}_{ij} - \mathbf{r}_{\rm rod}$. The free energy of the rod-network system has the form,

$$F(\mathbf{r}_{\rm rod}, \mathbf{l}_{\rm rod}) = -k_BT \ln Z(\mathbf{r}_{\rm rod}, \mathbf{l}_{\rm rod}) \quad (3)$$

To quantitatively examine the free energy experienced by the rod, the free energy change is defined as $\Delta F(\tilde{z}) = F(\tilde{z}) - F^{\rm min}(\tilde{z})$, where $\tilde{z}$ is the position of rod center as denoted by the coordinate in Fig. 4b, and $F^{\rm min}(\tilde{z})$ represents the minimum free energy along **z**-axis.

We calculate some typical profiles of $\Delta F$ as a function of $\tilde{z}/a_x$ for various $d/a_x$ at $L/a_x = 1.5$ and 2.0, corresponding respectively to the noncommensurate and commensurate rods (Fig. 4a). As illustrated by the schematic in Fig. 4b, $\Delta F$ reaches the maximum at the network cell center with $\tilde{z} = 0$, and the minimum at the network cell edge with $\tilde{z} = 0.5a_x$. Then, the height of the free energy barrier for the transition of the rod is $U_b = \Delta F^{\rm max}(\tilde{z}) - \Delta F^{\rm min}(\tilde{z})$, where $\Delta F^{\rm max}(\tilde{z})$ is the maximum of $\Delta F(\tilde{z})$. In view of the Kramers' escape theory[59], the noncommensurate rod facing a barrier of $U_b > k_BT$ (Fig. 4a) should possess hopping dynamics, consistent with simulation results in Fig. 2e. However, it is emphasized that the commensurate rod cannot give rise to hopping dynamics but a fast dynamics due to much lower barrier of $0 < U_b < k_BT$, as marked by the shaded region in Fig. 4a. To further assess how the free energy barrier $U_b$ relates to the dynamical regimes of hopping and fast dynamics, we systematically calculate $U_b$ for the same ranges of $d/a_x$ and $L/a_x$ with those in Fig. 2h, allowing us to construct $U_b$ landscape plotted by the colored contour map in the $d/a_x$-$L/a_x$ plane. As indicated by the color map in Fig. 2h, the landscape clarifies a strong dependence of dynamical regimes on $U_b$: hopping and trapped regimes take place within the range of $U_b > k_BT$, for noncommensurate rods; between two neighboring hopping regimes, there is a valley of $0 < U_b < k_BT$ for commensurate rods, corresponding to the fast dynamics. Moreover, the definite boundary of $U_b = k_BT$ is plotted, distinguishing the detailed ranges of fast dynamics around an integral multiple of $L/a_x$. This demonstrates that the fast dynamics is attributable to the anomalous "*faster*-than-expected" behavior of the sliding dynamics while the length of a thick rod reaches around an integral multiple of the network mesh size, and the emergence of sliding dynamics can be ascribed to such low free energy barrier, with which thermal noise may remarkably thrust the dynamical process. Moreover, the diffusion coefficients obtained in the

diffusive regime show nonmonotonic dependence on the rod length (Figs. 1c and 2d), indicating the changes of the free energy landscape.

The fairly smooth landscape reveals that the fast dynamics really follows the sliding dynamics characterized by the energy barrier of order of $k_BT$ and emerging predominantly in the diffusion of proteins along DNA[60,61]. Our simulations demonstrate that in such unique dynamics, MSD is simply proportional to time (Fickian), yet the DPDF is not Gaussian as should be expected of a classical random walk, bearing great resemblance to the "anomalous yet Brownian" diffusion as found in some crowded fluids containing colloidal particles, macromolecules and filaments[62,63].

Based on the calculations of the free energy landscape, we extend the theoretical analysis to microscopic dynamics to provide a refined picture of these dynamical regimes (see Supplementary Information IV for more details). Fundamentally, its microscopic dynamics can be described by the generalized Fokker-Planck equation[64],

$$D^\alpha G_s(\tilde{z},t) = \left[ \frac{\Delta F'(\tilde{z})}{\gamma} \frac{\partial}{\partial \tilde{z}} + \kappa \frac{\partial^2}{\partial \tilde{z}^2} \right] G_s(\tilde{z},t) \qquad (4)$$

where $\alpha$ is the anomalous coefficient. $\gamma = 2\pi\eta L/\ln(L/d)$ denotes the longitudinal friction coefficient of the rod, where $\eta$ is the viscosity, and $\gamma\kappa = k_BT$. The distribution of the total number of network cells traversed can be obtained in Fourier-Laplace space from the Montroll-Weiss equation[65],

$$S(k,s) = \frac{1 - \tilde{\psi}(s)}{s\left[1 - \hat{\phi}(k)\tilde{\psi}(s)\right]} \qquad (5)$$

where $S(k,s)$ is the Fourier-Laplace transform of $G_s(\tilde{z},t)$, $\hat{\phi}(k)$ and $\tilde{\psi}(s)$ are the respective Fourier and Laplace transforms of $\phi(z)$ and $\psi(t)$, denoting the waiting time and jump length distribution, respectively. The detailed forms of $G_s(\tilde{z},t)$ in the dynamical regimes are different, depending on $U_b$. Our following discussions are thereby based on the aforementioned regimes of $U_b$.

For high barriers of $U_b > k_BT$, we choose $\psi(t) = (1/\tau_{hop})\exp(-t/\tau_{hop})$, and $\phi(\tilde{z}) = CP(n)\delta(|\tilde{z}| - na_x)$ derived by Mel'nikov[66,67], where $\tau_{hop}$ is the characteristic hopping time, $P(y) = \exp(-4\beta U_0 y^2/a_x^2)$ gives the Boltzmann distribution of the particle at stationary state, $n$ is an integral number, and $C$ is the normalization constant such that $\sum_{n=0}^{\infty} CP(n) = 1$. Thus, one can deduce that for large time scales, it takes the form,

$$G_s(\tilde{z},t) = \left(\tau_{hop}/4\pi a_x^2 t\right)^{1/2} \exp\left(-\tau_{hop}\tilde{z}^2/4ta_x^2\right)\phi(\tilde{z}) \qquad (6)$$

where the exponential tails prevail. As the red curve in Fig. 4c, the plot of $G_s(\tilde{z},t)$ exhibits an identical shape as expected to the hopping dynamics shown in Figs. 1f and 2e.

For low barriers of $0 < U_b < k_BT$, a master equation is applied to respect the nonlocal nature of irregular peaks in $G_s(\tilde{z},t)$, which takes the form,

$$G_s(\tilde{z},t) = G_s(\tilde{z},0) + \int_0^t \sum_{\tilde{z}'} \omega(\tilde{z} - \tilde{z}') G_s(\tilde{z}',t')dt \qquad (7)$$

where the kernel $\omega$ has the standard Poisson random measure[68]. Then, the DPDF gives,

$$G_s(\tilde{z},t) = 4\exp(-2|\tilde{z}|/a_x)\exp(-2t/\tau_{hop})$$
$$\int_0^\infty dx \exp(-4x/a_x)I_0\left(\sqrt{\frac{8t(|\tilde{z}|+x)}{a_x\tau_{hop}}}\right)I_0\left(\sqrt{\frac{8tx}{a_x\tau_{hop}}}\right) \qquad (8)$$

where $I_0$ is the modified Bessel function of the first kind. Here we plot the blue curve of $G_s(\tilde{z},t)$ in the inset of Fig. 4c, demonstrating the random distribution of peaks resembling the sliding dynamics shown in Figs. 1g and 2f.

For no barrier of $U_b = 0$, the solution of Eq. 2 takes the Gaussian form

$$G_s(\tilde{z},t) = \left(\tau_0/4\pi a_x^2 t\right)^{1/2} \exp\left(-\tau_0\tilde{z}^2/4ta_x^2\right) \qquad (9)$$

where $\tau_0$ is the characteristic waiting time in solvents. As shown by the cyan curve in Fig. 4c the theoretical result is consistent with the simulated curve at the late stage (Fig. S7), reflecting the Gaussian distribution of Brownian dynamics.

## Discussion

Through conducting the experiments of single-particle tracking, performing molecular simulations and developing new theories, the transport dynamics of thick rods in macromolecular networks have been comprehensively investigated. We found that by tuning the rod length with respect to the averaged mesh size relatively fast longitudinal dynamics occurs once the rod length reaches around an integral multiple of the mesh size. We identified that the unconventionally fast transport follows sliding dynamics, which is demonstrated to be anomalous yet Brownian. We further gave the analytical expression of time-displacement distribution of sliding dynamics and clarified its physical relationship with hopping and Brownian dynamics. Our results revealed that the transition of these dynamical regimes is fundamentally attributed to the rod-length dependent free energy barrier originated predominantly from the entropic contribution due to the conformational penalty of strands deformed by thick rods, in contrast to the rigorous periodic potential. The findings might be of immediate interest to the optimal design of particle transport in various networks, biological or synthetic. As similar landscape of free energy is experienced by both passive and active rods in the confined environment of macromolecular networks, we speculate that the dependence of transport dynamics on size commensuration could be extended to the active case, suggesting the fundamental cornerstone for the further understanding of these nonequilibrium phenomena as well as the dissipative self-assembly of diverse building blocks[35,69].

## Methods

### Single-particle tracking of rods in synthetic networks

Full details are described in the Supplementary Information I and briefly introduced here. The PEGDA network was prepared under UV irradiation[37,70]. 4% wt/vol solution of PEGDA (20 kDa, JenKem Technology USA) was prepared in distilled water and 0.02 g/ml lithium phenyl-2,4,6-trimethylbenzoylphosphinate (LAP) photo initiator was added. The solution was cast on a 40 nm × 20 nm × 2 nm silicone slide with holes and cured under the UV light (wavelength 365 nm, 30 W) for 10 min. After the curing process, the PEGDA network was immersed under the distilled water for 48 h to remove the unreacted PEGDA monomers and allow the network to swell sufficiently. To obtain the diameter and length of Au-NRs, around 2.5 μL Au-NR sample was first diluted by dissolving in around 0.2 mL ethanol, sonicated for at least 20 min, and then dropped 3−6 μL solution on the transmission electron microscopy (TEM) copper grid. After evaporating any solvent on copper grids, we can obtain Au-NR images by TEM with 120 kV acceleration voltage. Au-NR sizes were measured from the TEM images by Fiji (ImageJ) software[40].

To determine the trajectories of Au-NR in PEGDA network, we filled the network with 30 μL Au-NR solution, which was kept for 10−15 min on an optical microscope stage (Olympus BX51) at room temperature before observing the Au-NR diffusion into the PEGDA network. For analyzing the trajectories of Au-NR in both water and

network, the images were taken by using dark field techniques[40], as schemed in Fig. 1a. For the water medium 400 images were recorded at a frequency of 20 Hz for 20 s time. For rods in the network, images were taken at a frequency of 10 Hz[11]. To detect the particles in the image stack, Crocker-Grier algorithm was used[44,45], and particles trajectories were then obtained by Nearest Neighbor Search tracker[45], as shown in Fig. S3. The tracking data was then exported in a CSV file format, and time-averaged MSDs, ensemble-averaged MSDs and diffusion coefficients were calculated, as shown in Figs. S4 and 1b, c, respectively.

## Coarse-grained molecular simulations

We simulate the transport of a rodlike particle in a cross-linked network using DPD simulations[49]. Full technical details on the simulation model are described in the Supplementary Information II and briefly introduced here. In the simulations, a bead represents a cluster of molecules, and a set of interacting beads are considered. The time evolution is governed by Newton's equations of motion. The force contains three parts: the conservative force, the dissipative force and the random force. In the conservative force, $a_{ij}$ represents a maximum repulsion between bead $i$ and bead $j$. Particularly, the interaction between like species $a_{ii}$ is set as 25[49]. Since all of these forces conserve momentum locally, hydrodynamic behavior emerges. The equations of motion are integrated in time with a modified velocity-Verlet algorithm. The factor $k_BT$ is taken as the characteristic energy scale. In our simulations, $k_BT = 1$. The characteristic time scale is then defined as $\tau = (mr_c^2/k_BT)^{1/2} = 1$. The remaining simulation parameter are $\gamma = 4.5$ and $\Delta t = 0.02\tau$ with a total bead number density of $\rho = 3r_c^{-3}$. To demonstrate the dynamics of a rod-like particle in the network, we choose a cubic box with dimensions of $42.76r_c \times 42.76r_c \times 42.76r_c$.

## Data availability

The data supporting the findings of this work are available within the paper and the Supplementary Information files. Source data are provided with this paper.

## Code availability

The code developed for this paper is made available at https://doi.org/10.5281/zenodo.10374051.

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

## Acknowledgements
We thank Pengyu Chen for stimulating discussion. We acknowledge support from National Natural Science Foundation of China (Grant Nos. 22025302, 21873053 and 22122201). L. T. Y. acknowledges financial support from Ministry of Science and Technology of China (Grant No. 2022YFA1203203) and State Key Laboratory of Chemical Engineering (No. SKL-ChE-23T01).

## Author contributions
L.T.Y. designed the research and conceived the project. X.Z., X.D., and L.T.Y. contributed to the development of the computational model and to its interpretation. X.D., X.Z., and L.T.Y. developed the analytical models and interpreted the results. H.A.M. and L.J. designed and performed the experiments. H.A.M. and X.D. collected the experimental data and carried out the analysis. L.T.Y., X.Z., and X.D. wrote the paper. L.G., W.C., W.W., Z.T., X.Q. and X.G. contributed to manuscript preparation and editing. All authors discussed the results and commented on the manuscript.

## Competing interests
The authors declare no competing interests.
