## [Peer Review File · Nature Communications]

REVIEWER COMMENTS

Reviewer #1 (Remarks to the Author):

Zhang and co-workers present an experimental study, coupled with simulations and theory, to investigate the diffusion of rod-like particles through a crosslinked polymer network. They focus on the regime in which the rod thickness is comparable to the mesh size of the network, which is a largely unexplored region of the phase space. They present intriguing results that show that motion of rods that have lengths that are integer multiples of the mesh size is significantly faster than for rods that lie between integer multiples. They explain their results as arising from the different propensities for the rods to undergo caging and hopping dynamics versus unencumbered longitudinal diffusion. Overall, I think the results are intriguing and the simulations seem to support the data. However I have some major concerns about how the data was analyzed (or omissions thereof) and interpreted and the lack of concrete comparison between simulation and experiment. There are also some of the claims that are made that demand further support. Below I expand on my major concerns relating to these overarching comments:

1. Measuring MSDs and determining diffusion coefficients

1a. Are the MSDs plotted 1D or 2D? Are they computed for the x or y direction? Or for the direction along the rod contour? The discussion in SI regarding longitudinal versus transverse diffusion makes me think that the MSD was evaluated along the direction of the rod length. But this would require advanced algorithms beyond the packages they use as the rod direction would need to be determined in each frame using principal component analysis or similar eigenvalue determination, and the displacement along that new axis determined. My guess is this is not the approach that was taken so how am I to understand the MSDs?

1b. The same question applies to the van Hove distributions - are these 1D or 2D and what direction are they computed for? How do they look for directions along and transverse to the contour?

1c. Following on a/b, from the images in the SI it doesn't appear that the orientation of the rods can be resolved so how are authors able to delineate between longitudinal and transverse motion (which seems essential to their claims)?

1d. The authors plot and discuss throughout the paper diffusion coefficients, however, many of the MSDs are subdiffusive which, by definition, prevents computation of a diffusion coefficient. Even the MSDs that appear to be roughly diffusive at long times only approach t^1 for less than half a decade in t , so the determination of D seems dubious. So how are diffusion coefficients determined? And how are errors

computed? (this should be in the caption). It seems more appropriate to discuss the anomalous scaling exponent if there is no real linear scaling of MSD with t .

1e. Following on d, all MSDs are subdiffusive at early lag times and then crossover to less diffusive. What is the relevance of the length and timescale associated with this crossover. This crossover is typically related to the confinement size or the characteristic size of the molecule. Moreover, the dependence of MSDs on L/a are different at short times versus long times so it seems different physics governs these two timescales. This important feature of the data and its interpretation warrants discussion.

2. Comparing experiments and simulation

2a. It is very difficult to see how well the simulation MSDs compare to experimental values because half of the simulation MSD data plotted is in the experimentally inaccessible regime (below 1 second) and is irrelevant to any discussion in the paper. The authors should plot the simulation data over the same timescale of the experimental data and include another x axis that uses seconds as the time instead of $\tau=0.017$ s.

2b. The same goes for the y-axis. The authors need to make clear what the units of r^2 are and how they relate to μm^2 used as the units in the experimental plot.

2c. The same critique in 1c holds in the simulation data. You cannot compute a diffusion coefficient from sublinear MSDs.

2d. To facilitate comparison, the authors should be consistent with color-coding so simulation and experimental data for the same L/a are the same color. Along the same lines, why are the simulated L/a values not identical to experimental values. It seems something you could easily control in simulations and would better facilitate comparison.

2e. The same critique in 2d holds for Fig 3a,b - why are the same L/a values not used? Also see 2a-d.

3. The authors refer to 'loops' around the rods at one point. What is meant by this? Are they referring to the size of the pores between the mesh? If so, these aren't really loops. Or is it something else?

4. The authors mention that the network is in theta solvent conditions. How was this confirmed? How would their results differ in good solvent conditions?

5. What are the interactions between the rod and mesh? I realize that simulations indicate that the interactions do not play a role but it is still an important experimental factor. Are there charge interactions or just steric? Is there sticking or is it no-slip boundary conditions? With the diameter being the same size as the mesh I would guess that there are more than simply steric interactions and that there is a lot of friction at the interfaces. However, there may also be charge attractions to drive the integer value rods to more effectively move through the mesh.

6. Referring to the dynamics the authors measure as 'speeding-up' dynamics is confusing and misleading. Are the rods actually moving faster than stokes-einstein predictions for a rigid rod? And faster than thin rods move? Or is it simply that a non-monotonic dependence is measured? Or do the authors mean that the dynamics transition from subdiffusion to diffusion? None of these are really 'speeding-up' and if anything I would argue that the caging/hopping is the anomalous 'slower-than-expected/predicted' behavior. But a much more clarification of these different dynamics and better terminology is needed.

7. I also do not understand the use of commensurate to refer to the rod as being an integer value of the mesh size.

Reviewer #2 (Remarks to the Author):

This paper presents experiment and simulations on the motion of thick rods in a crosslinked polymer network. The main result is that the diffusion constant of thick rods depends non-monotonically on the rod length. Rods with lengths commensurate with the mesh size move faster than rods which are not commensurate with the mesh size. These slower rods get trapped.

The paper presents some interesting results which could be published in Nature Communications but only after significant improvements. There are many ill-defined, non-standard terms which need to be clarified. I have marked the manuscript extensively with comments and questions that need to be addressed - too many to list here. I have also indicated places where grammar can be improved.

Reviewer #3 (Remarks to the Author):

Overall this is a very comprehensive study bringing together single particle tracking, simulations and theory to investigate nanorod diffusion in a network for the case where the rod diameter is similar to the mesh diameter. The results are important and the connection to biological systems brings this to another level. That said, there are improvements that need to be made before publication can be considered.

1) How does this study differ from Soft Condensed Matter 16 Apr 2023? The topic is very similar with some additional analysis in the present paper. However, several figures in the present paper are identical to those in the Soft Condensed Matter paper. Upon comparison of figures the SCM / NCOMMS maps as Fig1a/Fig 2a, Fig. 3a-b/Fig 2e-f/ Figure 2b/Fig 2h/ Fig 4/Fig4a-c. The SCM is an arXiv:2212.13341v2 paper so this is an editor's decision about reusing figures in an expanded study.

2) There are indeed few studies of nanorods with diameter on length scale of mesh. The authors should cite and describe wrt the present study the work by Rose, K.A. et al Macromolecules, (2022) in tetra-PEG gels. Similarly, for thin rods, please reference the classic deGennes, Brochard paper.

3) The term "accelerated" has specific meaning, namely the velocity increases/decreases with time. This paper suggests that rods with integer length diffuse faster than expected compared to non-integer rods. Diffusion is indeed faster but not accelerating as shown in the MSD plots.

4) Has the heterogeneity of the network been accounted for in the experimental studies? The variation of mesh size for free radical networks is likely much larger than 1.8nm claimed in the paper. This is relevant when observation of "hopping" is claimed in the paper. Hopping in the Rubinstein definition means moving from one mesh to another. Namely 21nm jump in this case. If the network has a wide range of mesh sizes (like modelled nicely later on), the hopping can be trapping of rods in a dense mesh, rattling around, and finally finding a more open mesh to jump into. How can the authors rule out that hopping isn't due to heterogeneity in the experiments?

5) The claim of hopping is interesting. As noted above, hopping from one mesh to the next (21nm) is the strict definition from Rubinstein's work. What resolution does the technique have? What exactly is the jump distance measured in Figure 1b?

6) Very nice to vary d/a_x to interaction strength between particle and strand. This is closer to real systems vs "ideal" strand-particle (brush) case.

7) Inclusion of mesh heterogeneity in mucus was very nice study. Connects model studies with real world biological systems.

8) The main issue is the disentanglement of the Soft Condensed Matter Paper and NCOMMS. Overall this is an outstanding study that I'd like to see published upon revision.

9) Please tone down the hype language. The paper is strong enough to stand on its own.

Manuscript ID: NCOMMS-23-36957

Manuscript title: Unconventionally fast transport through sliding dynamics of rodlike particles in macromolecular networks

Author(s): Xuanyu Zhang, Xiaobin Dai, Md Ahsan Habib, Ziyang Xu, Lijuan Gao, Zhongqiu Tang, Xianyu Qi, Xiangjun Gong, Lingxiang Jiang, and Li-Tang Yan

First of all, we would like to thank the reviewers for their thoughtful comments on our original manuscript. Our detailed responses to them are as follow:

Responses to Reviewer 1:

Zhang and co-workers present an experimental study, coupled with simulations and theory, to investigate the diffusion of rod-like particles through a crosslinked polymer network. They focus on the regime in which the rod thickness is comparable to the mesh size of the network, which is a largely unexplored region of the phase space. They present intriguing results that show that motion of rods that have lengths that are integer multiples of the mesh size is significantly faster than for rods that lie between integer multiples. They explain their results as arising from the different propensities for the rods to undergo caging and hopping dynamics versus unencumbered longitudinal diffusion. Overall, I think the results are intriguing and the simulations seem to support the data.

Reply: Thank you very much for the expert comments and suggestions!

However I have some major concerns about how the data was analyzed (or omissions thereof) and interpreted and the lack of concrete comparison between simulation and experiment. There are also some of the claims that are made that demand further support. Below I expand on my major concerns relating to these overarching comments:

Reply: We appreciate the comments! Thanks to the expert comments, in this version we have carried out a thorough revision to provide a clearer description regarding the data interpretation as well as the comparison between simulation and experiment. Furthermore, some new physical analysis has been supplied to make the presentation more solid. These changes, we believe, have made our manuscript more concise, transparent, and relevant to a broader audience. Thus, we hope the Editor and Reviewers like the revised work.

1. Measuring MSDs and determining diffusion coefficients

Reply: The expert comments are highly appreciated! Definitely considering MSDs and diffusion coefficients does be important for the understanding of the unique dynamic behaviors found in this work, and we are sorry for the possible ambiguity in the previous version. In the current version we have carried out a thorough revision to provide a clearer description regarding this aspect. In particular, some new physical analysis has been supplied to make the presentation more solid. Please let us explain them one-by-one in our replies to the following questions:

1a. Are the MSDs plotted 1D or 2D? Are they computed for the x or y direction? Or for the direction along the rod contour? The discussion in SI regarding longitudinal versus transverse diffusion makes me think that the MSD was evaluated along the direction of the rod length. But this would require advanced algorithms beyond the packages they use as the rod direction would need to be determined in each frame using principal component analysis or similar eigenvalue determination, and the displacement along that new axis determined. My guess is this is not the approach that was taken so how am I to understand the MSDs?

Reply: Thanks for the comment! Actually, in both experiments and simulations of this work the MSDs are plotted 1D, defined as $\langle \Delta z^2(t) \rangle = \langle [z(t+t_0) - z(t_0)]^2 \rangle$, where $z(t)$ is the displacement of the center of mass along the direction of the rod contour, as denoted by the green arrow in Fig.2a. Although this axis is readily determined in simulations, it does be identified through eigenvalue determination, which is based on principal component analysis algorithm, in the experiments. We are sorry for the omission of the introduction of this algorithm in the previous version.

Specifically, in the experiments the spots are detected and trajectories are obtained by the Trackpy software¹. We then apply the eigendecomposition algorithm to determine the direction of the rod^{2, 3}. For this purpose, the gyration tensor of the trajectory of a rod, is first calculated

$$\mathbf{S} = (S_{mn}) = \frac{1}{2N^2} \sum_{i=1}^N \sum_{j=1}^N (r_m^{(i)} - r_m^{(j)})(r_n^{(i)} - r_n^{(j)}) \quad (1)$$

where $r_m^{(i)}$ is the m^{th} Cartesian coordinate of the position vector of the i^{th} particle, and $N = t/\Delta t$ gives the total number of snapshots of a trajectory. The eigendecomposition of \mathbf{S} gives the major and minor components of the gyration tensor,

$$\mathbf{S} = \mathbf{V}\mathbf{\Lambda}\mathbf{V}^T \quad (2)$$

where $\Lambda = \text{diag}(\lambda_{\parallel}, \lambda_{\perp})$ is the eigenvalue matrix, $\lambda_{\parallel}, \lambda_{\perp}$ are respectively the characteristic parameters of the trajectory in longitudinal and transverse directions, and $\lambda_{\parallel} \geq \lambda_{\perp} > 0$. $\mathbf{V} = [\mathbf{v}_{\parallel}, \mathbf{v}_{\perp}]^T$ denotes the eigenvector matrix, and $\mathbf{v}_{\parallel}, \mathbf{v}_{\perp}$ are the unit vectors in longitudinal and transverse directions, respectively. Thus, the displacement along that new axis can be determined using this approach. More details regarding the application of this algorithm and the determination of the direction of the rod contour are presented in the following reply to Comment 1c.

In this revision, we have supplied the description regarding this aspect, which can be found, for example, from the 2nd to 4th line on Page 7 in the main text, and from the last 5 lines on Page 5 and first 8 lines on Page 6 in the Supplementary Information.

1b. The same question applies to the van Hove distributions - are these 1D or 2D and what direction are they computed for? How do they look for directions along and transverse to the contour?

Reply: Thanks for the comment! The van Hove distributions $G_s(z, t)$ are also plotted 1D, along the direction of the rod contour and corresponding to the MSDs. As stated above, the directions are identified based on eigenvalue determination, which is also delineated in our reply to the following comment, i.e., Comment 1c.

1c. Following on a/b, from the images in the SI it doesnt appear that the orientation of the rods can be resolved so how are authors able to delineate between longitudinal and transverse motion (which seems essential to their claims)?

Reply: Thanks for the careful comment! Although the orientation of the rods can hardly be identified due to the optical diffraction limit, their longitudinal and transverse motions can still be delineated by the eigendecomposition algorithm, which has been widely used in other systems with anisotropic particles^{2,3}. Based on this algorithm, below we present more details regarding the identification of the longitudinal and transverse motions of the rods in this work:

First, to apply this algorithm, the dynamics of rods should satisfy the condition that the rotational dynamics plays a trivial role during the observation period^{4,5}. To evaluate the rotational dynamics of a long and thick rod in our experiments, we perform a strict examination of the simulation and experimental results regarding the rotational dynamics of the rods in the present work. Fig. 1c in the main text and

Fig.R1 here present the representative trajectories for rods with different lengths corresponding to the hopping and sliding dynamics in the experiments and simulations, respectively. It can be identified that all trajectories form approximately straight lines across the sample rather than randomly diffusing in 2D, indicating that over the ranges of rod length and diameter explored in the present work, the rotational event takes place in an extremely low probability and thereby has little effect on the longitudinal motion of the rods in the macromolecular networks.

Fig. R1. Representative trajectories of rods with different length in simulations, where $d/a_x = 1.05$, and $L/a_x=2.6$ (a) and 3.0 (b).

Second, to consolidate the trivial role of the rotational dynamics during the observation period, we theoretically calculate the free energy barrier for the rod rotation in the macromolecular network and thereby estimate the waiting period for a rotational event. Full details and discussion on the rotational dynamics are described in Supplementary Information V. Briefly, the rotational diffusion of a rod is determined by two major parameters, one is the rotational diffusion coefficient D_r , and the other is the rotational free energy barrier ΔE_{rot} . For a typical rod used in this work, with diameter $d/a_x = 1.0$ and length $L/a_x = 2.0$, $D_r \approx 10.03s^{-1}$ and $\Delta E_{rot} \approx 10.3k_B T$. Using the Kramers' rate theory⁶, the waiting time for the rotational event can be estimated as,

$$t_{hop} = t_{r0} \exp(\Delta E_{rot} / k_B T) = \frac{1}{D_r} \exp(\Delta E_{rot} / k_B T) = 2.96 \times 10^3 \text{ s} \quad (3)$$

where $t_{r0} = 1/D_r$ is the waiting time for rotating at $\Delta E_{rot} = 0$ and can be estimated in polymer solution. This value is much larger than the observation period in each experiment (600s), corroborating that the rods adopt the longitudinal motion and the rotational dynamics does play a trivial role in their transport in these experimental systems.

Third, establishing the condition of restricted rotational dynamics allows us to apply the eigendecomposition algorithm to delineate between longitudinal and transverse motions, as detailed in our reply to Comment 1a.

Last, to further verify our results of longitudinal and transverse motions, we calculate the MSDs for different axial directions in both experiments and simulations. The MSDs along the major axis of the rods are much faster than those along the direction perpendicular to the major axis in both experiments (Fig. R2a) and simulations (Fig. R2b). More importantly, the transverse displacements are found to be highly subdiffusive, indicating strong transverse localization. The residual slight increase of MSD with time ($\text{MSD} \sim t^{0.2}$ empirically) likely reflects limited motions such as the thermal fluctuation of the polymer mesh⁴; in contrast, the parallel displacements are approximate to linear in elapsed time ($\text{MSD} \sim t^1$ empirically). This is consistent with the results of some previous works^{4,5} and corroborates the restricted transverse dynamics.

Fig. R2. $\langle \Delta r^2(t) \rangle$ in different axial directions for rods with different lengths in (a) experiments and (b) simulations. (//) parallel to the major axis; (\perp) perpendicular to the major axis.

In the revision, the above descriptions have been included, which can be found in, for example, from the 9th to 13th lines on Page 19, from the 16th to 20th lines on Page 19, the 1st line on Page 20, and the 9th line on Page 20, and the last 3 lines on Page 21 and the first 10 lines on Page 22 in the Supplementary Information (marked in red). Fig. R2a has also been included in it, as Fig. S12 in the Supplementary Information.

1d. The authors plot and discuss throughout the paper diffusion coefficients, however, many of the MSDs are subdiffusive which, by definition, prevents computation of a diffusion coefficient. Even the MSDs that appear to be roughly diffusive at long times

only approach t^1 for less than half a decade in t , so the determination of D seems dubious. So how are diffusion coefficients determined? And how are errors computed? (this should be in the caption). It seems more appropriate to discuss the anomalous scaling exponent if there is no real linear scaling of MSD with t .

Reply: The comment is appreciated! Actually, as shown in Figs.2 and 3 in the revision, all the MSDs of the simulations have reached the diffusive region with $\text{MSD} \sim t^1$, where the diffusion coefficients can be exactly determined. As shown in Fig.1d, the MSDs of most of the experiments have also reached the diffusive region, allowing us to exactly determine the diffusion coefficients. Only the cases at $L/a_x = 2.51$ and 2.63 are just approximate to the roughly diffusive region with a very short range in t^1 . In this revision, to exactly determine the diffusion coefficients at these both cases, we extend the total observation time to 10800s. Frankly, the measurement and data process within such a long period (10800s) is really challengeable; for example, the amount of the raw images is 108000, taking 216 GB for each length, and the computer memory for image processes, such as particle tracking, trajectory linking, ..., is almost energetically prohibited. Despite extremely challengeable, we still obtain the MSDs for these two samples of rods, which exhibit very similar trends with the previous results at the early stage and do evolve into the diffusive region with enough long range in t^1 at the later stage. It allows us to exactly determine their diffusion coefficients that are consistent with the previous results.

Fig. R3. Ensemble-averaged $\langle \Delta z^2(t) \rangle$ for different L/a_x with various measurement periods.

In the revision, the above process has been described, which can be referred to the last paragraph on Page 21 and the first paragraph on Page 22 in Supplementary Information.

1e. Following on d, all MSDs are subdiffusive at early lag times and then crossover to less diffusive. What is the relevance of the length and timescale associated with this crossover. This crossover is typically related to the confinement size or the characteristic size of the molecule. Moreover, the dependence of MSDs on L/a are different at short times versus long times so it seems different physics governs these two timescales. This important feature of the data and its interpretation warrants discussion.

Reply: We appreciate this comment! As definitely determined through theoretical analysis in section IV of the Supplementary Information and rightly pointed out by the Referee here, the subdiffusive behavior at early lag times is fundamentally attributable to the dynamical heterogeneity of the rod in the confined environment of the network^{7,8}. Physically, the length and timescale associated with this crossover are determined by the strength of the dynamic heterogeneity. On the one hand, the hopping dynamics possesses a stronger dynamical heterogeneity and thereby corresponds to a larger range of the subdiffusive regime than the sliding dynamics; on the other hand, for the same dynamical regime, a longer rod tends to give rise to a larger range of the subdiffusive regime due to the relative stronger confinement. The combination of these dynamic effects penalizes their relation to the thermodynamics at short times. In contrast, at long times all the MSDs evolve into the diffusive regimes with the same law $\text{MSD} \sim t^1$ over the crossovers, when the diffusive dynamics turns to being fully governed by the free energy landscape and thereby exhibits the rules as demonstrated by the length dependence of the diffusion coefficients (for example, Figs.1e and 2c) and established by the theoretical analysis in the main text.

In the revision, the above descriptions have been included, which can be found, for example, from the 9th to 11th lines on Page 15 in the main text (marked in red).

2. Comparing experiments and simulation

2a. It is very difficult to see how well the simulation MSDs compare to experimental values because half of the simulation MSD data plotted is in the experimentally inaccessible regime (below 1 second) and is irrelevant to any discussion in the paper. The authors should plot the simulation data over the same timescale of the experimental data and include another x axis that uses seconds as the time instead of $\tau=0.017$ s.

2b. The same goes for the y-axis. The authors need to make clear what the units of r^2 are and how they relate to μm^2 used as the units in the experimental plot.

Reply: Thanks for the comments! As these both comments refer to the same point, we reply to them together here.

Actually, the simulations as well as theoretical analysis are performed to explore the general rules and fundamental physics underlying the phenomena in this work. Both the simulation and experimental MSDs have the subdiffusive and diffusive regimes within almost the same ranges of the normalized rod length by the mesh size, allowing us to directly compare the rules of the dependence of the rod dynamic behaviors on the relation between the physical characters of the rods and networks. Indeed, both the simulations and analysis reproduce the rules demonstrated by the experimental phenomena, as revealed by the comparison between the results in Figs.1 and 2. In particular, when the parameters are normalized and dimensionless, the simulation results do be in good agreement with the experimental ones, as indicated in Fig.1e. These results corroborate that our simulations and analysis capture the fundamental physics underlying the experimental phenomena. As such, the direct comparison between the detailed values of the parameters, we believe, is principally unnecessary, as in many other systems focusing on the experimental phenomena and physics revealed by coarse-grained models in which the detailed scales of time and length can only be roughly estimated^{9,10}. Thus, we keep the axes of MSDs in simulations. According to other general cases^{8,11,12}, the ballistic regime in the much earlier stage of MSDs is also kept in the data of simulations to ensure the integrity.

2c. The same critique in 1c holds in the simulation data. You cannot compute a diffusion coefficient from sublinear MSDs.

Reply: We appreciate the careful comment! In the revision, the time scale of the simulation MSDs in, e.g., Fig. 2b, has been extended, ensuring an enough large range of the diffusive region for each MSD.

2d. To facilitate comparison, the authors should be consistent with color-coding so simulation and experimental data for the same L/a are the same color. Along the same lines, why are the simulated L/a values not identical to experimental values. It seems something you could easily control in simulations and would better facilitate comparison.

2e. The same critique in 2d holds for Fig 3a,b - why are the same L/a_x values not used?
Also see 2a-d.

Reply: Thanks for the comments! As these both comments refer to the same point, we reply to them together here.

In the revision, to facilitate comparison, the same color-coding has been adopted in the corresponding data between simulation and experiments, such as the MSDs in Figs. 1d, 2b and 3, where the lines at the similar L/a_x have the same color.

On the other hand, in DPD model used in the present work, a bead represents a fluid unit¹³ and the particle can only be built through packing the beads into a certain geometry^{14,15}. Thus, the size of a particle cannot be changed continuously, which causes a small deviation of size ratio (smaller than 0.1) in comparison to the corresponding value in experiments. However, such a tiny deviation cannot affect the trend as well as the fundamental physics at all.

3. The authors refer to ‘loops’ around the rods at one point. What is meant by this? Are they referring to the size of the pores between the mesh? If so, these aren't really loops. Or is it something else?

Reply: Thanks for the careful comment! A “loop” refers to a structural unit in a polymer network, which consists of a group of end-to-end connected polymer strands and usually exhibits circular shape, as schemed in the diagram of Fig.R4a. This concept has been generally used in the investigation of polymer networks, such as Fig.R4b from Ref. 7.

Fig. R4. (a) Schematic diagrams of hierarchical topologies of the polymer network, where the “loop” structure has been included. (b) Illustration of hopping dynamics of the nanoparticle in polymer network, and a network loop is highlighted in red⁷.

In the revision, the definition of “loop” has been added, which can be found, from the 14th to 15th lines on Page 11 in the main text (marked in red).

4. The authors mention that the network is in theta solvent conditions. How was this confirmed? How would their results differ in good solvent conditions?

Reply: The expert comment is highly appreciated! Frankly, after considering this comment carefully, we realize that in the previous version we misused “rods in the θ -solvent of network strands” to describe the entropic nature of the interaction between rods and network strands. Our basic consideration in this aspect was as follow:

In the DPD method, the interaction between species is controlled by varying the interaction parameter a_{ij} in the conservative force $\mathbf{F}_{ij}^C = a_{ij}(1 - r_{ij}/r_c)\hat{\mathbf{r}}_{ij}$. a_{ij} denotes the maximum repulsion between bead i and bead j , which has a linear relationship with Flory–Huggins χ parameter: $\chi_{ij} \approx (a_{ij} - a_{ii})/3.27$ ¹³. Usually, the interaction between like species a_{ii} is set as 25^{13,16}, meaning that there is no enthalpic preference^{13,17}. In this study, to bring out the entropic nature due to the conformational penalty of polymer strands deformed by a thick rod, the rod-strand interaction a_{rp} is set to be the same as that between like beads, that is, $a_{rp}=25$. This leads to $\chi_{rp} \approx 0$. On the other hand, according to the Flory-Huggins theory, the free energy change between these species is¹⁸

$$\Delta F = k_B T \left[\frac{\phi}{N_A} \ln \phi + \frac{1-\phi}{N_B} \ln(1-\phi) + \chi_{rp} \phi(1-\phi) \right] \quad (4)$$

Here the first two items in the bracket of the right-hand side represent the entropic contribution while the last one is the enthalpic contribution. By setting $\chi_{rp} \approx 0$, Eq.(4) is changed into

$$\Delta F \approx k_B T \left[\frac{\phi}{N_A} \ln \phi + \frac{1-\phi}{N_B} \ln(1-\phi) \right] \quad (5)$$

Clearly, only entropic items remain and the change of the free energy is governed by the entropy, capturing the physical nature of this system.

In the previous version, we tried to employ the solvent property to approximately describe such an entropic nature and turned to the concept of θ -solvent for the case of $a_{rp}=25$. However, after considering it carefully, we realize that it does be inappropriate and thereby have removed the description of θ -solvent in this revision.

In the revision, the description of this point has been revised, which can be found, for example, from the 14th to 15th lines on Page 8 and from the 11th to 18th lines on Page 8 in the Supplementary Information (marked in red).

5. What are the interactions between the rod and mesh? I realize that simulations indicate that the interactions do not play a role but it is still an important experimental factor. Are there charge interactions or just steric? Is there sticking or is it no-slip boundary conditions? With the diameter being the same size as the mesh I would guess that there are more than simply steric interactions and that there is a lot of friction at the interfaces. However, there may also be charge attractions to drive the integer value rods to more effectively move through the mesh.

Reply: Thanks for the comment! In this study, the thick rods, with their diameters comparable with or even larger than the averaged mesh size, are considered, which induce evident deformation of the network strands due to the *steric repulsion*. Fundamentally, it will cause remarkably entropic effect that can overwhelm other interactions and thereby dominates the change of the free energy of the system. Such an entropic nature has been captured in our simulation models, as detailed in our above reply to Comment 4. In particular, no charged species is included in the current work and thereby there is no charged interaction. Moreover, the friction, which generates the dissipation, has been exactly coupled in the DPD method though including the dissipative force¹³ (see Section II.1 of the Supplementary Information).

Furthermore, we actually have evaluated the influence of the enthalpy contributions through performing simulations by varying the particle-strand interaction parameter, a_{rp} . As demonstrated in Figs. 3a, c, S8 and their descriptions, the fast, longitudinal dynamics keeps once the rod length reaches around integral multiple of mesh size for both attractive ($a_{rp} = 20$) and repulsive ($a_{rp} = 30$) particle-strand interactions, resembling the results for the systems governed by almost purely entropic effects (i.e., Figs. 1 and 2). It underscores that the entropic contribution still dominates the behaviors over a wide range of the energetic interactions.

6. Referring to the dynamics the authors measure as ‘speeding-up’ dynamics is confusing and misleading. Are the rods actually moving faster than stokes-einstein predictions for a rigid rod? And faster than thin rods move? Or is it simply that a non-monotonic dependence is measured? Or do the authors mean that the dynamics transition from subdiffusion to diffusion? None of these are really ‘speeding-up’ and if anything I would argue that the caging/hopping is the anomalous ‘slower-than-expected/predicted’ behavior. But a much more clarification of these different dynamics and better terminology is needed.

Reply: We really appreciate this comment! Actually, the ‘speeding-up’ dynamics was used to describe the unconventionally *nonmonotonic dependence* of the diffusion coefficients of the thick rods on their length, where relatively fast longitudinal dynamics occurs once the rod length reaches around integral multiple of the mesh size, in striking contrast to the monotonic dependence of diffusion coefficients on the length of the thin rods. This, as revealed by both the simulations and the theoretical analysis, is attributable to the anomalous ‘*faster-than-expected/predicted*’ behavior of the *sliding* dynamics when the length of a thick rod reaches around integral multiple of the network mesh size.

After considering this aspect carefully, we agree with that the terminology of ‘speeding-up’ is inappropriate. In this revision, it is replaced by “fast” or “unconventionally fast” for a more exact description, which can be found, for example, from the 10th to 11th lines on Page 7, the 8th line on Page 8, and the 5th line on Page 9 (marked in red). This aspect has also been explained further in the revision, which can be found from the 14th to 16th lines on Page 6 (marked in red).

7. I also do not understand the use of commensurate to refer to the rod as being an integer value of the mesh size.

Reply: Thanks for the careful comment! To eliminate the confusion and ensure concise description, in the revision, we give the definition of “commensurate length” which definitely related it to “integral multiple of mesh size”. This can be found in, for example, from the 10th to 11th lines on Page 7 (marked in red).

Responses to Reviewer 2:

This paper presents experiment and simulations on the motion of thick rods in a crosslinked polymer network. The main result is that the diffusion constant of thick rods depends non-monotonically on the rod length. Rods with lengths commensurate with the mesh size move faster than rods which are not commensurate with the mesh size. These slower rods get trapped.

The paper presents some interesting results which could be published in Nature Communications but only after significant improvements. There are many ill-defined, non-standard terms which need to be clarified. I have marked the manuscript extensively with comments and questions that need to be addressed - too many to list here. I have also indicated places where grammar can be improved.

Reply: Thank you very much for the expert comments and suggestions!

In this revision, we consider the comments and address the issues with great care, including a thorough checking of the writing to avoid any grammar/typesetting error. In particular, we list all the points marked in the manuscript and respond to them one-by-one here. Moreover, to facilitate your checking, we also give the same response or reply at each position listed in the manuscript *attached to this letter*.

Position: Page 2 Line 8, the

Comment: a

Reply: Thanks for the comment! In the revision, “the” has been revised into “a”, which can be found in the 7th line on Page 2 in the revision (marked in red).

Position: Page 2 Line 9 Mark1, commensuration-governed speeding-up

Comment: this statement has no meaning - one can say D is nonmonotonic with L, ie longer rods are faster than shorter one but speeding-up is vague and not meaningful.

Reply: Thanks for the comment! As our reply to Comment 6 of Reviewer 1, the ‘speeding-up’ dynamics was used to describe the nonmonotonic dependence of the diffusion coefficients of the thick rods on their length. After considering the comments carefully, we agree with that the terminology of ‘speeding-up’ is inappropriate. In this revision, it is replaced by ‘fast’ or ‘unconventionally fast’ for a more exact description, which can be found, for example, in the 9th line on Page 2 (marked in red). This aspect has also been explained further in the revision, which can be found from the 14th to 16th lines on Page 6 (marked in red).

Position: Page 2 Line 9 Mark 2, and in

Comment: which is in...

Reply: We appreciate the comment! In the revision, “and in” has been revised into “which is in”, which can be found in the 9th line on Page 2 in the revision (marked in red).

Position: Page 2 Line 12 Mark 1, anomalous yet Brownian.

Comment: it can't be both - if its anomalous its not Brownian - Brownian has a precise meaning and it does not include anything anomalous – reword.

Reply: Thanks for the comment! Actually, “anomalous yet Brownian” is a defined concept which was proposed and defined in the work of Ref.19 and has been widely adopted to describe the corresponding dynamic behaviors in diverse systems¹⁹⁻²¹. Generally, this concept defines a class of anomalous dynamics in which the MSD is Fickian while the displacement probability distribution function (DPDF) is non-Gaussian^{19,20}. In the present work, our results demonstrate that the sliding dynamics satisfies these features and thereby belongs to the scope of this class of anomalous dynamics. Thus, we use this concept to demonstrate the physics and universal nature of the sliding dynamics.

In the revision, this concept has been introduced, which can be found in the 2nd paragraph on Page 15 (marked in red). Moreover, double quotation has been added to this term, that is, “anomalous yet Brownian”, to avoid any ambiguity, which can be found, for example, in the 12th line on Page 2 (marked in red).

Position: Page 2 Line 12 Mark 2, a

Comment: delete “a”.

Reply: Thanks for the careful comment! In the revision, “a” has been deleted, which can be found in the 12th line on Page 2 (marked in red).

Position: Page 2 Line 13, the sliding dynamics

Comment: do authors mean that MSD is faster than t^1 power - is that what is meant by sliding dynamics. Its not a common way to describe motion so it must be clarified - I don't see any region in figure 1c that shows that MSD goes like t^x where $x > 1$, so what does sliding dynamics mean to authors - this must be clarified here and in paper.

Reply: Thanks for the comment! Actually, the sliding dynamics is a general concept defining the approximately linear diffusion of an object confined in the one-dimensional (1D) space^{22,23}. Such a unique dynamic behavior has been detected in diverse systems²²⁻²⁴, and is characterized by the low energy barrier of order of $k_B T$ ²³. In this work, we find that the unconventionally fast dynamics of the thick rods confined in the network cells of a macromolecular network follows the sliding dynamics. Furthermore, our theoretical analysis gives the analytical expression of the DPDF of this dynamics and clarifies its physical relationship with hopping and Brownian dynamics.

In the revision, this concept has been described, which can be found in the 2nd paragraph on Page 15 (marked in red).

Position: Page 3 Line 15, energy

Comment: delete “energy”

Reply: Thanks for the careful comment! In the revision, “energy” has been deleted, which can be found in the 15th line on Page 3 (marked in red).

Position: Page 4 Line 6, trivial

Comment: minor

Reply: Thanks for the comment! In the revision, “trivial” has been replaced with “minor”, which can be found in the 6th line on Page 4 (marked in red).

Position: Page 4 The last line, The result has important merits

Comment: Here we show that the translation diffusion is nonmonotonic and depend on ... thin rods. We derived an analytic experssion for the time-...

Reply: Thanks for the comment! In the revision, the corresponding sentence “The result has important merits” has been revised into “The merits of the results are listed as follow”, which can be found in the last line on Page 4 (marked in red).

Position: Page 5 Line 2, around integral

Comment: around an integral

Reply: We appreciate the comment! In the revision, “around integral” has been replaced with “around an integral” , which can be found, for example, in the 2nd line on Page 5 (marked in red).

Position: Page 5 Line 4, rods

Comment: add reference

Reply: Thanks for the careful comment! In the revision, the related reference has been added, that is Ref.25 in the main text, which can be found in the 4th line on Page 5 (marked in red).

Position: Page 5 Line 7, It allows new

Comment: This work provides new principles ... synthetic.

Reply: Thanks for the comment! In the revision, “It allows new...” has been revised into “This work provides new...”, which can be found in the 7th line on Page 5 (marked in red).

Position: Page 5 Line 11, in the

Comment: in a

Reply: Thanks for the careful comment! In the revision, “in the” has been revised into “in a”, which can be found in the 11th line on Page 5 (marked in red).

Position: Page 5 Line 12, where diameters

Comment: where the diameters

Reply: The comment is appreciated! In the revision, “where diameters” has been revised into “where the diameters”, which can be found in the 12th line on Page 5 (marked in red).

Position: Page 5 Line 13, a_x

Comment: size a_x of the network.

Reply: Thanks for the comment! In the revision, “size a_x ” has been revised into “size a_x of the network”, which can be found in the 13th line on Page 5 (marked in red).

Position: Page 5 Line 14, PEGDA

Comment: The PEGDA

Reply: Thanks for the careful comments! In the revision, “PEGDA” has been revised into “The PEGDA”, which can be found in the 14th line on Page 5 (marked in red).

Position: Page 5 Line 19, the

Comment: delete “the”

Reply: Thanks for the comment! In the revision, “the” has been deleted, which can be found in the 19th line on Page 5 (marked in red).

Position: Page 6 Line 7, The representative

Comment: Representative .

Reply: Thanks for the comment! In the revision, “The representative” has been revised into “Representative”, which can be found in the 7th line on Page 6 (marked in red).

Position: Page 6 Line 11, “jump” motion

Comment: usually this is referred to as 'hopping'

Reply: We appreciate the comment! In the revision, “jump” has been revised into “hopping”, which can be found in the 11th line on Page 6 (marked in red).

Position: Page 6 Line 12 Mark 1, until $t = 300\text{s}$

Comment: for ~ 300 ps and

Reply: Thanks for the comment! In the revision, “until $t=300\text{s}$ ” has been revised into “for ~ 300 s”, which can be found in the 12th line on Page 6 (marked in red).

Position: Page 6 Line 12 Mark 2, jumps

Comment: hops

Reply: Thanks for the comment! In the revision, “jumps” has been revised into “hops”, which can be found in the 12th line on Page 6 (marked in red).

Position: Page 6 Line 13, the continuous and speeding-up

Comment: Continous. not sure what speeding-up means - clarify/explain - all one can say from this figure is that motion is continuous - how fast them move can only be discussed after one show diffusion vs L/a_x .

Reply: Thanks for the comments! The “speeding-up” has been deleted and only “continuous” remains, which can be found in the 13th line on Page 6 (marked in red).

Position: Page 6 Line 14 Mark 1, scenery

Comment: definitely not scerney - maybe situation or case.

It appears that over this time the motion is not diffusive but ballistic, ie $r(t) \sim t$ not $t^{1/2}$. Is this true or is it really diffusion $r(t) \sim \sqrt{t}$. if the former it must be stated clearly.

Reply: The expert comment is appreciated! In the revision, “scenery” has been replaced with “situation”. Actually, the ballistic regime can’t be identified in experiments because it occurs within extremely early and short time scale. Indeed, over this time, the motion has evolved into the diffusive regime, as confirmed by the $\text{MSDs} \sim t^1$ in Fig. 1d.

Position: Page 6 Line 14 Mark 2, diffusivity of the thick rod nonmonotonically depend on L

Comment: this statement should follow not precede the discussion of diffusion const vs L as only 2 images are presented, not statistics of motion.

Any statements of non-monotonic behavior can only follow presentation of D vs L.

Reply: Thanks for the comment! We really agree with this point! In the revision, the “where the diffusivity of the thick rod nonmonotonically depend on L ” have been changed into “where different dynamic behaviors appear”, which can be found from the 13th to 14th lines on Page 6 (marked in red).

Position: Page 6 Line 15, is increase of

Comment: ref 20 doesn't show nonmonotonic dependence - do you mean 'unlike in thin rods - ref 20 showed that D decreased L^{-1} as expected. Clarify

Reply: Thanks for the careful comment! In the revision, this sentence has been revised into “such a change of the dependence of diffusivity on rod length for the thick rod is distinct from the monotonic situation for the thin rod”, which can be found from the 14th to 16th lines on Page 6 (marked in red).

Position: Page 6 Line 18: trajectories for rods with different lengths

Comment: see discussion on figure caption - top result show extended ballistic motion ($r(t)$ vs t) - is this true, Is this an anomaly, what do other trajectories for $L=2.02$ show.

Position: Page 30 Line 7, $L/a_x: 2.02 \pm 0.18$,

Comment: the result for 2.02 appears to be super diffusive ($r(t)$ increasing faster than $t^{1/2}$) - more ballistic ($r(t) \sim t$) - it would be useful to show several trajectories for the same L - is this superdiffusive behavior for $L=2.02$ typical or an anomaly? It doesn't seem to show up in average msd.

Could this be an issue with the network having inhomogeneities -

Reply: The expert comments are highly appreciated! As these both comments refer to the same point, we reply to them together here.

After considering this issue with great care, we find that the choice of the trajectory at $L/a_x=2.02$ in the previous version does be unrepresentative. Such a special trajectory seems to be extended ballistic and doesn't show up the behavior in the ensemble-averaged MSD where the rod motion has indeed reached the diffusive region with $MSD \sim t^1$, as confirmed in Fig.1d. Please let us explain it in more detail as follow:

As show in Fig.S4 in the Supplementary Information, although the ensemble-average MSD at each length presents the statistical behavior of the rod diffusion, the time-averaged MSDs measured for different particles with the same L/a_x can exhibit deviations to the corresponding ensemble-averaged value, which, actually, is general and has been demonstrated in various systems^{26,27}. In Fig. R5, we present experimental results for the time-averaged MSDs and their

ensemble-averaged MSD for the rod at $L/a_x=2.02$. Some of the representative trajectories are also presented in Fig. R6, corresponding to the typical time-averaged MSDs as marked by the bold lines with different colors in Fig. R5. One can find that, the time-averaged MSD for the trajectory used in the previous version, i.e., the red line, shows superdiffusive behavior to a certain extent, while the blue line exhibits approximately subdiffusive behavior and corresponds to a shorter trajectory. In this revision, we choose a more representative trajectory whose time-averaged MSD has much smaller deviation to the ensemble-averaged MSD, as demonstrated by the green line in Fig.R5.

Fig. R5. Time-averaged (gray lines) and ensemble-averaged (purple line) MSDs $\langle \Delta z^2(t) \rangle$ plotted against time on the log–log scale, at $L/a_x = 2.02$. The red, green and blue lines correspond to the trajectories shown in Fig.R6, respectively.

Fig. R6. Trajectories of rods at $L/a_x=2.02$, which correspond to the lines of MSDs shown in Fig. R5, respectively.

In the revision, the trajectory at $L/a_x=2.02$ has been replaced with the one stated above, which can be found in Fig.1c.

Position: Page 6 Line 19, jump

Comment: hop, what is speeding-up motion?

Reply: Thanks for the comment! As stated above, “jump and speeding-up motions” has been changed into “hop and continuous motions”, which can be found from the 18th to 19th lines on Page 6 (marked in red)

Position: Page 6 Line 21, jump

Comment: hop

Reply: Thanks for the careful comment! As stated above, “jump” has been replaced with “hop”, which can be found in the 21st line on Page 6 (marked in red)

Position: Page 6 Line 22, unconventional speeding-up dynamics of the

Comment: what does this mean - is this in reference to the 2.0 and 2.97 rods or the 2.51 rods. Are authors claiming that the hopping results in overall faster diffusion (larger displacements with time)??

Reply: The careful comment is appreciated! Yes, it is in reference to the 2.0 and 2.97 rods. To avoid any confusion, the sentence “indicating unconventional speeding-up dynamics of the thick rods in the network” has been changed into “indicating unconventional dynamics for the thick rods with lengths of integral multiple mesh size” in the revision, which can be found from the 21st to 22nd lines on Page 6 (marked in red).

Position: Page 7 Line 3, MSDs for a set of L/a_x ,

Comment: how many t_o is this data averaged over - as noted above - a plot of $\Delta r^2(t)$ for individual starting time t_o would be usefully (say 20-40 starting time t_o , well separated) would be very informative.

Reply: We appreciate the comment! In the present work, about 30 individual starting time t_o , with separated time 10s, are adopted for each time-averaged MSD.

In the revision, the above description has been included, which can be found from the 19th to 20st lines on Page 4 in the Supplementary Information.

Position: Page 7 Line 8, that the speeding-up

Comment: don't use speeding-up - dynamics is faster or diffusion is larger but speeding-up is not appropriate.

Reply: Thanks for the comment! In the revision, “speeding-up” has been revised into “unconventionally fast”, which can be found in the 10th line on Page 7 (marked in red).

Position: Page 7 Line 18, of speeding-up

Comment: of the dynamics - remove 'speeding-up'

Reply: Thanks for the comment! In the revision, “speeding-up” has been removed, which can be found in the 11st line on Page 7 (marked in red).

Position: Page 8 Line 6, coarse-grained molecular simulations

Comment (1): need to say in main text that simulations are using a DPD model - it appears from discussion that there is no extra constraint on the beads to avoid chain segments from cutting each other.

Reply: Thanks for the careful comment! “coarse-grained molecular simulations” has been replaced with “dissipative particle dynamics (DPD)”, which can be found from the 7th to 8th lines on Page 8 (marked in red).

To avoid chain segments from cutting each other, we adopted a general approach through setting $K_b = 64k_B T$ in the harmonic spring potential $U_{\text{bond}} = K_b[(r - b)/r_c]^2$ for the bonds of the network strands, which has been widely used to tackle this issue in the DPD simulations¹³, which can be found from the 4th to 7th lines on Page 9 in the Supplementary Information.

Position: Page 8 Line 6, coarse-grained molecular simulations

Comment (2): What is the concentration of the beads - is the system at melt density or is there a lot of empty space with is modeled as an implicit solvent.

Reply: Thanks for the comment! In DPD, there is explicit solvent instead of implicit solvent. According to this method¹³, the bead number density is usually set $\rho = 3$, which has been given in the 5th lines on Page 20 (marked in red).

Position: Page 8 Line 6, coarse-grained molecular simulations

Comment (3): As interactions are purely repulsive this is not a model for a theta-solvent. - this is incorrect. The interactions being all of the same strength does

not make it theta-like.

Reply: The expert comment is highly appreciated! Although the same question has been responded in our reply to Comment 4 of Reviewer 1, please let us repeat it as follow:

Frankly, after considering this comment carefully, we realize that in the previous version we misused “rods in the θ -solvent of network strands” to describe the entropic nature of the interaction between rods and network strands. Our basic consideration in this aspect was as follow:

In the DPD method, the interaction between species is controlled by varying the interaction parameter a_{ij} in the conservative force $\mathbf{F}_{ij}^C = a_{ij}(1 - r_{ij}/r_c)\hat{\mathbf{r}}_{ij}$. a_{ij} denotes the maximum repulsion between bead i and bead j , which has a linear relationship with Flory–Huggins χ parameter: $\chi_{ij} \approx (a_{ij} - a_{ii})/3.27$ ¹³. Usually, the interaction between like species a_{ii} is set as 25^{13,16}, meaning that there is no enthalpic preference^{13,17}. In this study, to bring out the entropic nature due to the conformational penalty of polymer strands deformed by a thick rod, the rod-strand interaction a_{rp} is set to be the same as that between like beads, that is, $a_{rp}=25$. This leads to $\chi_{rp} \approx 0$. On the other hand, according to the Flory-Huggins theory, the free energy change between these species is¹⁸

$$\Delta F = k_B T \left[\frac{\phi}{N_A} \ln \phi + \frac{1-\phi}{N_B} \ln (1-\phi) + \chi_{rp} \phi (1-\phi) \right] \quad (4)$$

Here the first two items in the bracket of the right-hand side represent the entropic contribution while the last one is the enthalpic contribution. By setting $\chi_{rp} \approx 0$, Eq.(4) is changed into

$$\Delta F \approx k_B T \left[\frac{\phi}{N_A} \ln \phi + \frac{1-\phi}{N_B} \ln (1-\phi) \right] \quad (5)$$

Clearly, only entropic items remain and the change of the free energy is governed by the entropy, capturing the physical nature of this system.

In the previous version, we tried to employ the solvent property to approximately describe such an entropic nature and turned to the concept of θ -solvent for the case of $a_{rp}=25$. However, after considering it carefully, we realize that it does be inappropriate and thereby have removed the description of θ -solvent in this revision.

In the revision, the description of this point has been revised, which can be found from the 14th to 15th lines on Page 8 in the main text (marked in red) and from the 14th to 18th lines on Page 8 in the Supplementary Information.

Position: Page 8 Line 9, is taken to be a hexa

Comment: explain why simulation is on a perfectly ordered network and experiment is on a randomly crosslinked network which surely is much more inhomogenous than the simulation. Explain how this could affect the results.

Reply: Thanks for the expert comment! The predominant goal of this work is to capture the general principle as well as fundamental physics behind the phenomena, and therefore the value of each characteristic of the network is set within the reasonable scale corresponding to the general cases. In particular, a relatively ordered network, in which the polymer strands actually exhibit various conformations due to the thermal noise, facilitates us to build the theoretical models and definitely clarify the fundamental physics. Despite of this, to verify the reproducibility of the simulation results in the experimental conditions, we have performed additional simulations to examine the effect of the networks with inhomogenous mesh sizes, which actually has been detailed in the section of “Entropic effect examined through evaluating diverse factors” in the main text and is briefly introduced as follow:

To consider the impact of the polydispersity of mesh sizes on the dynamical behaviors of the rodlike particles, we first quantify the distribution of network mesh sizes with the coefficient of variation (CV), defined as $CV = \sigma_{ax}/a_x$, where σ_{ax} is the standard deviation of the mesh size. On one hand, using the rheology test, we find that CV of the macromolecular networks used in the current experiments is less than 0.1, which drops well in the range of the simulations where the macromolecular networks with $CV = 0.3$ and 0.7 are used. As shown in Fig. 3 (b) and (d), the simulation results can basically fall to the physical principles revealed based on the regular network, underscoring that this neat model can be applied to delineate the dynamical behaviors in the networks with a reasonable range of heterogeneity, especially for the *thick* rods which can generate remarkable structural deformation and thereby significantly impair the influence of the mesh polydispersity.

Position: Page 8 Line 9, 50

Comment: superscript

Reply: Thanks for the careful comment! In the revision, the number has been changed into superscript, which can be found in the 10th line on Page 8 (marked in red).

Position: Page 8 Line 14, network mesh size is fixed at

Comment: network mesh size is not fixed - network fluctuates so one has an average

mesh size, its not fixed.

Reply: Thanks for the comment! In the revision, “the network mesh size is fixed at” has been revised into “the average network mesh size is at around”, which can be found from the 15th to 16th lines on Page 8 (marked in red).

Position: Page 8 Line 20, take ballistic

Comment: move ballistically

Reply: Thanks for the comment! In the revision, “take ballistic motions” has been revised into “move ballistically”, which can be found in the last line on Page 8 (marked in red).

Position: Page 9 Line 4, speeding-up

Comment: faster diffusion - don't use speeding-up

Reply: The comment is appreciated! In the revision, “speeding-up” has been replaced with “fast”, which can be found in the 5th line on Page 9 (marked in red).

Position: Page 9 Line 10, speeding-up

Comment: ordinary diffusion or diffusive motion

Reply: Thanks for the expert comment! In the revision, “speeding-up” has been revised into “fast”, which can be found in the 10th line on Page 9 (marked in red).

Position: Page 10 Line 3, without any peaks

Comment: what does this mean

Reply: Thanks for the comment! The displacement probability distribution function (DPDF) of Brownian dynamics exhibits a Gaussian distribution, which has no harmonic peak. To avoid confusion, in the revision “without any peaks” has been revised into “with Gaussian distribution”, which can be found in the 4th line on Page 10 (marked in red).

Position: Page 10 Line 21, it turns to trapped dynamics

Comment: trapped dynamics has not meaning - subdiffusive regime before crossing over to diffusive - is this what is meant.

Reply: Thanks for the expert comment! “Trapped dynamics” means that a particle undergoes a remarkable confinement so that it can only fluctuate slightly around its equilibrium position, which, in fact, is one of the typical dynamic states for a particle

confined in the network environment^{7,8}.

In the revision, to ensure a clear description, “it turns to trapped dynamics” has been revised into “it turns to the trapped dynamics characterized by slight fluctuation around its equilibrium position”, which can be found from the 17th to 18th lines on Page 10 (marked in red).

Position: Page 10 Line 26, speeding-up

Comment: faster dynamics

Position: Page 10 Line 27, speeding-up

Comment: don't use speeding-up

Reply: The comments are appreciated! As these both comments refer to the same point, we reply to them together here.

After considering them carefully, in the revision the first “speeding-up” has been replaced with “unconventionally fast” and the second one with “the faster”, which can be found in the first two lines on Page 10 (marked in red) in the revision.

Position: Page 11 Line 13, loop

Comment: segments between crosslinks? is this what is meant - what is a loop

Reply: Thanks for the comment! Although the same question has been responded in our reply to Comment 3 of Reviewer 1, please let us repeat it as follow:

A “loop” refers to a structural unit in a polymer network, which consists of a group of end-to-end connected polymer strands and usually exhibits circular shape, as schemed in the diagram of Fig.R4a. This concept has been generally used in the investigation of polymer networks, such as Fig.R4b from Ref. 7.

In the revision, the definition of “loop” has been added, which can be found, from the 14th to 15th lines on Page 11 (marked in red).

Position: Page 11 Line 17, nature

Comment: nature of what?

Reply: We appreciate the careful comment! In the revision, “nature” has been revised into “nature of the length-dependent, nonmonotonic diffusion dynamics of thick rods”, which can be found from the 19th to 20th lines on Page 11 (marked in red).

Position: Page 11 Line 18, setting d/a_x at a typical value but change...

Comment: fixing, fixing d/a_x and varying the particle-...

Reply: Thanks for the comment! In the revision, “setting d/a_x at a typical value but change...” has been revised into “fixing d/a_x and varying the ...”, which can be found in the 21st line on Page 11 (marked in red).

Position: Page 12 Line 1, speeding-up

Comment: faster ..dynamics continues once ..

Reply: Thanks for the comment! In the revision, “speeding-up longitudinal dynamics keeps once” has been revised into “faster longitudinal dynamics continues once”, which can be found from the 3rd to 4th lines on Page 12 (marked in red).

Position: Page 12 Line 2, attractive ($arp = 20$)

Comment: all the interactions are repulsive - $arp=20$ is not attractive - its just less repulsive than the value 25 between.

it is well know that changing the relative repulsive strength does little or nothing to the system. The fact that nothing changed completely expected as the enthalpic interactions are really not changed by varying arp in a purely repulsive model. This subsection should be removed as it does not show that authors claim - only by adding truly attraction can one effect the enthalpic interactions. one only has excluded volume interaction when potential is purely repulsive.

Reply: Thanks for the comment! Actually, the detailed information on the interaction parameter a_{ij} in the DPD model, especially its relation to a general theory in polymer physics, i.e., the Flory-Huggins theory, has been delineated in our above reply to the following comment:

“Position: Page 8 Line 6, coarse-grained molecular simulations

Comment (3): As interactions are purely repulsive this is not a model for a theta-solvent. - this is incorrect. The interactions being all of the same strength does not make it theta-like.”

Here, please let us briefly repeat it and thereby respond to this comment as follow:

In the DPD method, the interaction between species is controlled by varying the interaction parameter a_{ij} in the conservative force $\mathbf{F}_{ij}^C = a_{ij}(1 - r_{ij}/r_c)\hat{\mathbf{r}}_{ij}$. a_{ij} denotes the maximum repulsion between bead i and bead j , which has a linear relationship with Flory–Huggins χ parameter: $\chi_{ij} \approx (a_{ij} - a_{ii})/3.27$ ¹³. Usually, the interaction between like species a_{ii} is set as 25^{13,16}, meaning that there is no enthalpic preference^{13,17}. In this study, to bring out the entropic nature due to the conformational penalty of polymer strands deformed by a thick rod, the rod-strand interaction a_{rp} is set to be the

same as that between like beads, that is, $a_{rp}=25$. This leads to $\chi_{rp} \approx 0$, capturing the physical nature of this system where the change of the free energy is predominantly governed by the entropy. Moreover, when $a_{rp} < 25$, for example, $a_{rp} = 20$, $\chi_{rp} < 0$, which corresponds to the attractive enthalpic interaction between the rods and the polymer network, especially considering the bath of other beads with stronger repulsive interactions (that is, the interaction parameters are equal with or larger than 25).

In the revision, the description of this point has been revised, which can be found from the 14th to 15th lines on Page 8 in the main text (marked in red) and from the 14th to 18th lines on Page 8 in Supplementary information.

Position: Page 12 Line 9, causes the polydispersity of molecular structures of the networks

Comment: this is repeat of what was stated above - remove repetitions.

Reply: Thanks for the comment! Actually, the discussion in this section is distinct from that in the above sections of the main text, because here we turn to addressing the issue of how the *inhomogenous mesh sizes* of the network affect the length-dependent, nonmonotonic diffusion dynamics of thick rods, in contrast to the above discussion based on the *relatively regular* network. This point has been delineated in our above reply to the following comment:

“*Position: Page 8 Line 9, is taken to be a hexa*

Comment: explain why simulation is on a perfectly ordered network and experiment is on a randomly crosslinked network which surely is much more inhomogenous than the simulation. Explain how this could affect the results.”

Here, please let us briefly repeat it and thereby respond to this comment as well as the following two comments, as follow:

The predominant goal of this work is to capture the general principle as well as fundamental physics behind the phenomena, and therefore the value of each characteristic of the network is set within the reasonable scale corresponding to the general cases. In particular, a relatively ordered network, in which the polymer strands actually exhibit various conformations due to the thermal noise, facilitates us to build the theoretical models and definitely clarify the fundamental physics. Despite of this, to verify the reproducibility of the simulation results in the experimental environments, we have performed additional simulations to examine the effect of the networks with *inhomogenous* mesh sizes, which is present and discussed in this section.

Position: Page 12 Line 20, results of the macromolecular networks, whose polydispersity approximates to those of typical biological networks

Comment: what does this mean - simulations are on a perfect network with all segment lengths the same - experiments are on much more disordered systems - so it incorrect to claim that the simulations have a polydispersity that approximated the bio networks. They may have similar ave mesh size but distributions in the 2 are surely very different.

Reply: Thanks for the comment! Following the above reply, to consider the impact of the polydispersity of mesh sizes on the dynamical behaviors of the rodlike particles, we first quantify the distribution of network mesh sizes with the coefficient of variation (CV), defined as $CV = \sigma_{a_x}/a_x$, where σ_{a_x} is the standard deviation of the mesh size. On one hand, using the rheology test, we find that CV of the macromolecular networks used in the current experiments is less than 0.1, which drops well in the range of the simulations where the macromolecular networks have been *built with random mesh size distribution* of CV = 0.3 and 0.7 (as demonstrated in Fig.R7), mimicking some real macromolecular networks²⁸⁻³⁰.

Fig. R7. Distribution of the mesh size a_x for networks with different polydispersity, where CV = 0.1 (experimental network in the present work, red bar), 0.3 (green bar) and 0.7 (blue bar), respectively.

Position: Page 13 Line 3, effect induced by the thick rods accounts for the unconventional dynamic behaviors, as the influence of the mesh polydispersity is significantly impaired upon the remarkable structural deformation.

Comment: not proven, speculation

Reply: Thanks for the comment! Following the above two replies, as shown in Fig. 3 (b) and (d), the simulation results can basically fall to the physical principles revealed based on the regular network. Therefore, we believe it establishes that the neat model can be applied to delineate the dynamical behaviors in the networks with a reasonable range of heterogeneity, especially for the *thick* rods which can generate remarkable structural deformation and thereby significantly impair the influence of the mesh polydispersity.

Position: Page 12 Line 18, speeding-up

Comment: change

Reply: We appreciate the careful comment! In the revision, “speeding-up” has been revised into “fast”, which can be found in the 18th line on Page 12 in the main text (marked in red).

Position: Page 19 Line 11, We simulate the transport of a rodlike particle....

Comment: why is this repeated here and in SI - doesn't need to be in both. need to say explicitly this is a DPD model -

Reply: The comment is appreciated! We have changed the description of the DPD method in the section of Methods and Supplementary Information (SI). These revisions are aimed at ensuring that the main text stands alone without over-relying on the SI, thus making the methodology more transparent and concise.

Position: Page 30 Line 8, the sliding, hopping and sliding dynamics

Comment: what is sliding dynamics - there is no region where msd goes faster than t.

Reply: Thanks for the comment! Actually, this point has been delineated in our above reply to the following comment:

“Position: Page 2 Line 13, the sliding dynamics

Comment: do authors mean that MSD is faster than t^1 power - is that what is meant by sliding dynamics. Its not a common way to describe motion so it must be clarified - I don't see any region in figure 1c that shows that MSD goes like t^x where $x > 1$, so what does sliding dynamics mean to authors - this must be clarified here and in paper.”

Here, please let us briefly repeat it and thereby respond to this comment as follow:

The sliding dynamics is a general concept defining the approximately linear

diffusion of an object confined in the one-dimensional (1D) space^{22,23}. Such a unique dynamic behavior has been detected in diverse systems²²⁻²⁴, and is characterized by the low energy barrier of order of $k_B T$ ²³. In this work, we find that the unconventionally fast dynamics of the thick rods confined in the network cells of a macromolecular network follows the sliding dynamics, as demonstrated in Figs.1 and 2 where MSD is simply proportional to time t , yet the DPDF is non-Gaussian. Furthermore, our theoretical analysis gives the analytical expression of the DPDF of this class of dynamics and clarifies its physical relationship with hopping and Brownian dynamics.

As the concept regarding the sliding dynamics occurs logically in the later section of the main text, here “sliding” is replaced with “fast” for a more exact description, which can be found in the 8th line on Page 30 in the main text in the revision (marked in red).

Position: Page 30 Line 23 Mark 1, speeding-up

Comment: what does this mean - simple diffusion?

Reply: Thanks for the comment! Actually, this point has been delineated in our above reply to the following comment:

“Position: Page 2 Line 9 Mark1, commensuration-governed speeding-up

Comment: this statement has no meaning - one can say D is nonmonotonic with L , ie longer rods are faster than shorter one but speeding-up is vague and not meaningful.”

Here, please let us briefly repeat it and thereby respond to this comment as follow:

The ‘speeding-up’ dynamics was used to describe the nonmonotonic dependence of the diffusion coefficients of the thick rods on their length. After considering the comments carefully, we agree with that the terminology of ‘speeding-up’ is inappropriate. In this revision, it is replaced by ‘unconventionally fast’ or ‘fast’ for a more exact description, which can be found, for example, in from the 10th to 11th lines on Page 7 (marked in red). This aspect has also been explained further in the revision, which can be found from the 14th to 16th lines on Page 6 (marked in red).

Position: Page 30 Line 23 Mark 2, trapped

Comment: what is trapped dynamics - no rods are trapped, they may have a subdiffusive regime before diffusing but they are not trapped - trapped dynamics has no meaning

Reply: We appreciate the expert comment! Actually, this point has been delineated in our above reply to the following comment:

“Position: Page 10 Line 21, it turns to trapped dynamics

Comment: trapped dynamics has not meaning - subdiffusive regime before crossing over to diffusive - is this what is meant.”

Here, please let us briefly repeat it and thereby respond to this comment as follow:

“Trapped dynamics” means that a particle undergoes a remarkable confinement so that it can only fluctuate slightly around its equilibrium position, which, in fact, is one of typical dynamic states for a particle confined in the network environment^{7,8}.

Responses to Reviewer 3:

Overall this is a very comprehensive study bringing together single particle tracking, simulations and theory to investigate nanorod diffusion in a network for the case where the rod diameter is similar to the mesh diameter. The results are important and the connection to biological systems brings this to another level. That said, there are improvements that need to be made before publication can be considered.

Reply: Thank you very much for the expert comments and suggestions!

1) How does this study differ from Soft Condensed Matter 16 Apr 2023? The topic is very similar with some additional analysis in the present paper. However, several figures in the present paper are identical to those in the Soft Condensed Matter paper. Upon comparison of figures the SCM / NCOMMS maps as Fig1a/Fig 2a, Fig. 3a-b/Fig 2e-f/ Figure 2b/Fig 2h/ Fig 4/Fig4a-c. The SCM is an arXiv:2212.13341v2 paper so this an editors decision about reusing figures in an expanded study.

Reply: The comment is highly appreciated! Actually, the current paper is the submitted version of the preprint in the arXiv with No. 2212.13341, and we can confirm the fact that this is the *only* submitted version for publication of that preprint. When we were preparing the work for submission and publication, we uploaded the previous version to Arxiv preprint platform, but forgot to update it to the latest version and to note this matter upon the last submission. *We are so sorry for the carelessness!!*

When we just received the decision letter of the last submission, we *immediately* updated it to the submitted version as arXiv:2212.13341v3 preprint on Arxiv. Now, it

has been updated again to the latest version corresponding to this revision as arXiv:2212.13341v4 preprint. We will *certainly and timely* update its information for the publication in the future.

Anyway, if there is still any conflict with the submission to Nature Communications, we will withdraw the preprint from the Arxiv platform.

Moreover, the above message has been included in the Cover Letter to this submission.

2) There are indeed few studies of nanorods with diameter on length scale of mesh. The authors should cite and describe wrt the present study the work by Rose, K.A. et al *Macromolecules*, (2022) in tetra-PEG gels. Similarly, for thin rods, please reference the classic deGennes, Brochard paper.

Reply: Thanks for the comment! These important literatures have been included in the revision, as Ref. [21] and [71] in the main text.

3) The term "accelerated" has specific meaning, namely the velocity increases/decreases with time. This paper suggests that rods with integer length diffuse faster than expected compared to non-integer rods. Diffusion is indeed faster but not accelerating as shown in the MSD plots.

Reply: The expert comment is highly appreciated! After considering this aspect carefully, we fully agree with that the terminology of 'accelerated' is inappropriate. In this revision, it is replaced with "fast" or "unconventionally fast" for a more exact description, which can be found, for example, in from the 10th to 11th lines on Page 7 (marked in red). This aspect has also been explained further in the revision, which can be found from the 14th to 16th lines on Page 6 (marked in red).

4) Has the heterogeneity of the network been accounted for in the experimental studies? The variation of mesh size for free radical networks is likely much larger than 1.8nm claimed in the paper. This is relevant when observation of "hopping" is claimed in the paper. Hopping in the Rubinstein definition means moving from one mesh to another. Namely 21nm jump in this case. If the network has a wide range of mesh sizes (like modelled nicely later on), the hopping can be trapping of rods in a dense mesh, rattling around, and finally finding an more open mesh to jump into. How can the authors rule out that hopping isn't due to heterogeneity in the experiments?

Reply: We really appreciate this expert comment! Actually, the definition of the hopping of a rod in the present work is the same as that defined in the Rubinstein's work⁷, that is, the center of mass of a particle jumps from one mesh to another. In the present work, the averaged mesh size is measured as $21.0 \pm 1.8 \text{ nm}$, offering a relatively neat system to definitely determine the length-dependent, nonmonotonic diffusion dynamics of the thick rods. Indeed, the jump distance is just around the integral multiple of mesh size, as clarified in Fig.1f, which consolidates that hopping isn't due to heterogeneity in the experiments. Of course, the hopping dynamics may be impaired for a macromolecular network with wider distribution. However, we believe that it can be captured if the heterogeneity of the network is not too strong.

5) The claim of hopping is interesting. As noted above, hopping from one mesh to the next (21nm) is the strict definition from Rubinstein's work. What resolution does the technique have? What exactly is the jump distance measured in Figure 1b?

Reply: Thanks for the comment! The frame rate of the tracking video is set as 10Hz, so that the low threshold of the temporal resolution becomes 10^{-1} s . Furthermore, we use the Crocker-Grier algorithm³¹ in particle tracking experiments, by which the particle's position can be localized with *subpixel* accuracy through taking the average position of not only the brightest pixel but also its neighboring pixels, weighted by brightness, so that the low threshold of the spatial resolution can reach about 5nm, allowing us to definitely determine the distance shorter than an averaged mesh size. Furthermore, the jump distance is measured as around 21nm, as also confirmed by the DPDF shown in Fig.1f.

6) Very nice to vary d/λ to interaction strength between particle and strand. This is closer to real systems vs "ideal" strand-particle (brush) case.

Reply: Thanks for the expert comment!

7) Inclusion of mesh heterogeneity in mucus was very nice study. Connects model studies with real world biological systems.

Reply: The expert comment is really appreciated, thanks!

8) The main issue is the disentanglement of the Soft Condensed Matter Paper and NCOMMS. Overall this is an outstanding study that I'd like to see published upon revision.

Reply: Thanks for the expert comment and suggestion! We are sorry again for this issue! As stated in the reply to Comment 1, we believe that this issue has been solved appropriately.

9) Please tone down the hype language. The paper is strong enough to stand on its own.

Reply: Thank you very much for the expert comments and kind suggestions!

In this revision, we consider this aspect carefully, and have revised the description for a more strict and appropriate presentation. Thanks again!

With these statements, we sincerely hope that the paper is now suitable for publication.

Thanks!

Li-Tang Yan

Lingxiang Jiang

References

- [1] Allan, D. B., et al., *Soft-Matter/Trackpy: Trackpy v0.5.0; Zenodo*.
- [2] Parsa, S., Guasto, J. S., Kishore, M., et al. *Phys. Fluids* **23**, 043302 (2011).
- [3] Pumir, A. & Wilkinson, M. *New J. Phys.* **13**, 093030 (2011).
- [4] Tsang, B., Dell, Z. E., Jiang, L., et al. *Proc. Natl. Acad. Sci. U.S.A.* **114**, 3322 (2017).
- [5] Fakhri, N., Wessel, A. D., Willms, C., et al. *Science* **344**, 1031 (2014).
- [6] Cohen, A. E. *Phys. Rev. Lett.* **94**, 118102 (2005).
- [7] Cai, L. H., Panyukov, S. & Rubinstein, M. *Macromolecules* **48**, 847-862 (2015).
- [8] Xu, Z., Dai, X., Bu, X., et al. *ACS Nano* **15**, 4608-4616 (2021).
- [9] Eloul, S., Poon, W. C., Farago, O., and Frenkel, D. *Phys. Rev. Lett.* **124**, 188001 (2020).
- [10] Mladek, B. M., Fornleitner, J., et al. *Phys. Rev. Lett.* **108**, 268301 (2012).
- [11] Larini, L., Ottochian, A., De Michele, C., et al. *Nat. Phys.*, **4**, 42-45 (2008).
- [12] Dai, X., Zhang, X., Gao, L., et al. *Nat. Commun.* **13**, 4094 (2022).
- [13] Groot R. D. and Warren P. B. *J. Chem. Phys.* **107**, 4423 (1997).

- [14] Yan, L. T., Popp, N., Ghosh, S. K., and Boker, A. *ACS Nano* **4**, 913-920 (2010).
- [15] Alexeev, A., Uspal, W. E., and Balazs, A. C. *ACS Nano* **2**, 1117-1122 (2008).
- [16] Shin, J. M., Kim, Y., Yun, H., et al. *ACS Nano* **11**, 2133-2142 (2017).
- [17] Khani, S., Jamali, S., Boromand, et al. *Soft Matter* **11**, 6881-6892 (2015).
- [18] Rubinstein, M. & Colby, R. *Polymer Physics* (Oxford University Press, 2003).
- [19] Wang, B., Anthony, S. M., et al. *Proc. Natl Acad. Sci. U. S. A.* **106**, 15160-15164 (2009).
- [20] B. Wang, J. Kuo, C. Bae & S. Granick, *Nat. Mater.* **11**, 481 (2012).
- [21] J. Kim, C. Kim & B. J. Sung, *Phys. Rev. Lett.* **110**, 047801 (2013).
- [22] Loverdo, C., Benichou, O., Voituriez, R., et al. *Phys. Rev. Lett.* **102**, 188101 (2009).
- [23] Gorman, J. & Greene, E. C. *Nat. Struct. Mol. Biol.* **15**, 768-774 (2008).
- [24] Shang, C., Xiong, Z., Liu, S. & Yu, W. *Macromolecules* **55**, 3711-3722 (2022).
- [25] Wang J., O'Connor T. C., Grest G. S., et al. *Macromolecules* **54**, 7051-7060 (2021).
- [26] Rose, K. A., Gogotsi, N., Galarraga, J. H., et al. *Macromolecules*, **55**, 8514-8523 (2022).
- [27] Crocker, J. C., Valentine, M. T., Weeks, E. R., et al. *Phys. Rev. Lett.*, **85**, 888 (2000).
- [28] Yu, M., Wang, J., Yang, Y., et al. *Nano Lett.* **16**, 7176 (2016).
- [29] Lai, S. K., Wang, Y.-Y., Hida, K., et al. *Proc. Natl. Acad. Sci. U.S.A.* **107**, 598-603 (2010).
- [30] Figureueroa-Morales, N., Dominguez-Rubio, L., Ott, T. L. et al. *Sci. Rep.* **9**, 9713 (2019).
- [31] Crocker J. C. & Grier D. G. *J. Colloid Interface Sci.* **179**, 298-310 (1996).

REVIEWERS' COMMENTS

Reviewer #1 (Remarks to the Author):

The authors have addressed all of my concerns satisfactorily. I recommend the revised version for publication. But I would read through carefully and correct for grammatical issues and poor English that still exist in certain places.

Reviewer #2 (Remarks to the Author):

The authors have addressed my previous comments. After addressing one more point, I recommend publication.

I could not find what the strand length (number of beads) for the simulated networks. Total number of bonds were given but not the total number of beads and strand length. Also in reply the authors indicate that the network is in an explicit solvent, however I could not find the number of solvent beads and the number of network beads. If there are solvent beads, then what is the density of the network, total density is set =3.

Reviewer #3 (Remarks to the Author):

The reviewer has met enough comments to warrant publication.

Manuscript ID: NCOMMS-23-36957A

Manuscript title: Unconventionally fast transport through sliding dynamics of rodlike particles in macromolecular networks

Author(s): Xuanyu Zhang, Xiaobin Dai, Md Ahsan Habib, Ziyang Xu, Lijuan Gao, Zhongqiu Tang, Xianyu Qi, Xiangjun Gong, Lingxiang Jiang, and Li-Tang Yan

First of all, we would like to thank the reviewers for their thoughtful comments on our manuscript. Our detailed responses to them are as follow:

Responses to Reviewer 1:

The authors have addressed all of my concerns satisfactorily. I recommend the revised version for publication. But I would read through carefully and correct for grammatical issues and poor English that still exist in certain places.

Reply: Thank you very much for the expert comments and suggestions!

The English and writing of this work have been checked more carefully, to avoid any possible grammatical issue in this revision.

Responses to Reviewer 2:

The authors have addressed my previous comments. After addressing one more point, I recommend publication.

I could not find what the strand length (number of beads) for the simulated networks. Total number of bonds were given but not the total number of beads and strand length. Also in reply the authors indicate that the network is in an explicit solvent, however I could not find the number of solvent beads and the number of network beads. If there are solvent beads, then what is the density of the network, total density is set =3.

Reply: Thank you very much for the expert comments and suggestions!

The number of beads in a strand is 6, corresponding to the strand length of about $3.35r_c$. The total number of beads is 234549, leading to the total density $\rho = 3r_c^{-3}$, which has been given in the 7th line on Page 20 of the revision. In these beads, the

number of network beads is 42180 and the number of the solvent beads is 192369. Thus, the density of the network beads is about $0.54r_c^{-3}$.

In this revision, the above description has been added in the section of “II. Details of simulation models (1. Coarse-grained molecular simulations)” in the Supplementary Information, which can be found in the last 4 lines on Page 9 of the Supplementary Information.

Responses to Reviewer 3:

The reviewer has met enough comments to warrant publication.

Reply: Thank you very much for the expert comments and suggestions!

With these statements, we sincerely hope that the paper is now suitable for publication.

Thanks!

Li-Tang Yan

Lingxiang Jiang